# ARE LANGUAGE MODEL OUTPUTS CALIBRATED IN NUMERIC CONTEXTS?

## ABSTRACT

Some information is factual (e.g., "Paris is in France"), whereas other information is probabilistic (e.g., "the coin flip will be a $[Heads/Tails]$."). We believe that good Language Models (LMs) should understand and reflect this nuance. Our work investigates this by testing if LMs' output probabilities are *calibrated* to their textual contexts. We define model "calibration" as the degree to which the output probabilities of candidate tokens are aligned with the relative likelihood that should be inferred from the given context. For example, if the context concerns two equally likely options (e.g., heads or tails for a fair coin), the output probabilities should reflect this. Likewise, context that concerns non-uniformly likely events (e.g., rolling a six with a die) should also be appropriately captured with proportionate output probabilities. We find that even in simple settings the best LMs (1) are poorly calibrated, and (2) have systematic biases (e.g., preferred colors and sensitivities to word orderings). For example, `gpt-4o-mini` often picks the first of two options presented in the prompt regardless of the options' implied likelihood, whereas `Llama-3.1-8B` picks the second. Our other consistent finding is mode-collapse: Instruction-tuned models often over-allocate probability mass on a single option. These systematic biases introduce non-intuitive model behavior, making models harder for users to understand.

## 1 INTRODUCTION

We investigate the extent to which language model (LM) output probabilities are calibrated to the numeric content of their contexts. Consider the contexts below:

(1) From **17** *blue* marbles and **99** *red* marbles, Tommy reached blindly into a bag and grabbed a marble that was the color $[blue/red]$

(2) From **98** *blue* marbles and **99** *red* marbles, Tommy reached blindly into a bag and grabbed a marble that was the color $[blue/red]$

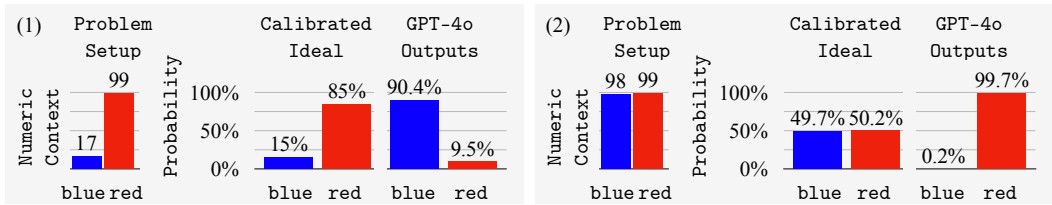

Figure 1: **Models produce un-calibrated results**. Inputting Examples 1 and 2 to `gpt-4o` different, uncalibrated behaviors arise in the model probabilities. For (1), `gpt-4o` over-weights the first option listed in the prompt, predicting *blue* with 90.4% when 15% is appropriate. For (2), `gpt-4o` over-weights the option with a higher number beyond the calibrated ratio, predicting *red* with 99.7% when 50.2% is appropriate. We term the first behavior `Pick First` and the second `Pick Higher`.

We believe the probabilities for the next generated token should be calibrated to the relevant numeric content, capturing some of the nuance of probabilistic information. E.g., in the examples above, the

probability of an LM continuing with the *bracketed tokens* should be proportional to the respective **bolded** values. In Example 1, the probability of *blue* should be approximately $\frac{1}{5}$th the probability of *red*. We also evaluated how models behave when prompted to sample from distributions, like below:

(3)   Sampling from a uniform distribution with support from **2** inclusive to **5** exclusive, I sampled the number *[0/1/2/3 . . . 9]*

In Example 3, the tokens *2, 3, 4* should each have a probability of $1/3$, while other digits should have probabilities close to 0. In other words, when the target output is based directly on probabilities discernible from the context, randomness should arise from the model probabilities and the sampling algorithm, rather than from arbitrary factors within the context.

> **What Do We Mean By Calibration?**
>
> LMs output probabilities are *calibrated* to their contexts if the probabilities of relevant tokens correspond to the numeric content implicitly or explicitly defined by their contexts. If tokens $t_1, t_2, \ldots t_n$ are indicated by context $C$ to have probabilities $P = p_1, p_2 \ldots p_n$, a calibrated LM $m$ outputs corresponding probabilities $\pi_i = m(t_i|c)$:
>
> $$p_i \propto \pi_i, \quad i \in 1, 2 \ldots n.$$
>
> We measure calibration by the distance between $P$ and $\Pi = \pi_1, \pi_2, \ldots \pi_n$, Section 3.2.

Stepping back, our motivation is drawn from the fact that while pre-training endows language models with a wide range of capabilities (Brown et al., 2020a; Wei et al., 2022), further training can increase the utility of existing models by encouraging them to follow instructions (Wei et al., 2021; Sanh et al., 2021) or match human preferences (Ouyang et al., 2022) rather than human behavior implicit in natural human text. Part of this project is ensuring that a model's overt reasoning process aligns with its internal reasoning process (Lanham et al., 2023). This idea is also critical for model explanation methods (Chen et al., 2023): counterfactual simulatability requires that an explanation captures the "reason" a model arrived at an answer. Along the same lines, for contexts that call for contextual calibration, e.g., situations where the context suggests probabilities over the next tokens, the most direct solution is for models to yield exactly those probabilities. Kusner et al. (2017) also studies probability differences in the setting of counterfactual fairness.

There are large-scale efforts for models to not reproduce endemic online biases (Bai et al., 2022; Rudinger et al., 2018); we test if models are calibrated to the numeric content in context rather than other patterns. Such biases are common with people, like the first-mentioned bias (Van Koevering & Kleinberg, 2024) where the first option listed is favored. LLMs have also been shown to exhibit biases concerning option order in multiple-choice questions (Pezeshkpour & Hruschka, 2023).

> **Contributions: Models Are *Not* Calibrated**
>
> Though different models have different error patterns, no model is well-calibrated. Moreover, there are some shared themes across models and model families:
>
> 1. Mode-collapse: Instruction-tuning tends to improve the allocation of probability mass upon valid options, but instruction-tuning also leads to decreased entropy in model outputs. This can both improve (such as in Example 1) or reduce (such as in Example 3) calibration.
>
> 2. Models have systematic tendencies: Many models show preferences for certain tokens (such as numbers or colors), and the word order can also influence which tokens the models favor.

## 2   RELATED WORK

**Models struggle to calibrate internal uncertainty with textual outputs.**   A related but separate line of work has studied the calibration in neural networks in terms of their ability to match their prediction confidence with prediction accuracy: a network with $0.8$ confidence on each prediction should classify $80\%$ of examples correctly (Guo et al., 2017; Minderer et al., 2021). Another line of work investigates how well the model outputs align with uncertainty (Yona et al., 2024; Kumar

et al., 2024; Zhou et al., 2023; Lin et al., 2022), linguistic calibration (Mielke et al., 2022). Yona et al. (2024) report that models struggle to exhibit their internal uncertainty with text, while Kumar et al. (2024) measure how well model confidence is aligned internally (model probabilities) and externally (via eliciting model to choose among a Likert scale.) In our work, we look at the calibration from the perspective of agreement between the probability of predicted outputs and the contextually defined likelihood of outputs in the text. We find that models often are *certain* when they should be properly *uncertain*; mode-collapse arises both via supervised fine-tuning and reward-based fine-tuning (O'Mahony et al., 2024; Janus, 2022). Zhou et al. (2024) likewise find that RLHF biases models towards certainty, leading to an over-generation of strengtheners (e.g., "I am certain...") over weakeners (e.g., "Maybe...").

**Recent work investigates if language models can output random results.** Van Koevering & Kleinberg (2024) shows that single coin flip predictions are biased towards heads and otherwise demonstrate a first-mentioned bias. Silva (2024) shows that large models can 1) correctly describe Matching Pennies, a game popular in game theory, and 2) code optimal behavior. Hopkins et al. (2023), work closely related to our own, examines whether language models can induce uniform distributions in the 0 to 1 interval.

## 3 EXPERIMENTAL DESIGN

We run two experiments that evaluate if model probabilities are calibrated to the numeric values either explicit or implicit in context, Section 3.4 (`Distributions`) and Section 3.5 (`Probabilities`). See Appendix appendix A for more information on reproducing these experiments.

### 3.1 PROBLEM SETUP

Each problem assumes a context $C$ to be continued by a relevant token among $t_1, t_2, \ldots t_n = T \subset V$ where $V$ is the full vocabulary. Each token $t_i$ is associated with a probability $p_i$ forming a distribution, $P = p_1, p_2, \ldots p_n$. In Example 1, $T$ is $\{red, blue\}$ with $P = \{red : 0.75, blue : 0.25\}$. For a model $m$, we define its probabilities over these words as $\Pi = \pi_1, \pi_2, \ldots$ where $\pi_i \approx m(t_i|C)$, with one addendum. We sum the probabilities of all tokenizations for a given word, which can include capitalization and spaces: $\pi_i = \sum_{s \in \text{Tokenizations}(t_i)} m(s|C)$. E.g., we sum probabilities for "red", "Red", "_red", and "_Red."

### 3.2 METRICS AND EVALUATION

Our primary metrics below help us ask how well a model calibrates to numeric information in context.

**Probability Mass (PM)**: Calibration, the relative probability mass on tokens in $T$, only matters if the **PM** that falls upon $T$ is sufficient, $\text{PM}(T) \doteq \sum_{t : T} \pi_t$. If $\text{PM}(T)$ for a model is low (say, 0.30), this indicates that the model is not capturing the intended relationship between the context and $T$. However, when PM is high (say, 0.75), we can start to meaningfully ask questions about how probability mass is allocated among, say, *red* and *blue*. While our metric for measuring calibration (introduced below) would also capture when the PM is too low, separately measuring this information makes it easier to understand and compare model behavior.

**Relative Entropy (RE):** To detect mode collapse, we measure **RE**, the ratio between entropy H of model probabilities and the calibrated probabilities, $\text{H}(\Pi)/\text{H}(P)$. An RE less than one implies that the probability mass is overly concentrated.

**Wasserstein Distance (WD)**: To measure **calibration**, we use **WD**. WD (Kantorovich, 1939) captures the movement between one distribution (or set of values) and another.[1]

---

[1] Kullback–Leibler (KL) divergence is also a natural choice. However, a number of the target distributions we have include 0 values where KL would be undefined. We also compute Mean-Squared Error (MSE) but don't find a material difference in the findings so only report the WD.

## 3.3 Models

We test open-source models that have both base and instruction-tuned versions: `Mistral-7B-v0.1`, `Mistral-7B-v0.3`, `Mixtral-8x7B-v0.1`, `Yi-1.5-9B`, `Yi-1.5-34B`, `Meta-Llama-3.1-8B`, `gemma-2-9b`, `gemma-2-27b` (Jiang et al., 2023; 2024; Young et al., 2024; Dubey et al., 2024; Team et al., 2024); and four proprietary models, `gpt-3.5`, `gpt-4-turbo`, `gpt-4o-mini`, and `gpt-4o` (Brown et al., 2020b; Achiam et al., 2023; OpenAI, 2024). We prompt instruction-tuned models to be calibrated. We refer to the base, unprompted version of the models as "base" and the instruction-tuned, prompted models as "chat." Model keys are listed in Appendix Table 4.

## 3.4 Experiment: Distributions

This experiment tests if models can produce a uniform distribution in their output probabilities. Example 3, repeated below, is representative of the instances in this dataset:

> (3) Sampling from a uniform distribution with support from **2** inclusive to **5** exclusive, I sampled the number $[0/1/2/3 \ldots 9]$

We test across different numeric ranges, range inclusivities, and templates. The numbers scale from 10 to 903 and we use five different templates; there are 4500 unique instances. In some of our analyses, we group problem ranges and inclusivities that support the same tokens: For example, [12, 15) and [92, 95) are grouped. There are 615 such groups.

From here, the question is if the distribution of the next token is calibrated to the uniform distribution $\Pi^C = \mathcal{U}(2,5)$. As noted above, we report WD, which for this example would be: $\mathrm{WD}(\Pi^T, \mathcal{U}(2,5)) = \mathrm{WD}(\Pi^T, [0, 0, 1/3, 1/3, 1/3, 0, \ldots, 0]$, where $T$ includes digits (tokens) 0 to 9 and $\mathcal{U}(2,5)$ covers all digits (with 0 for unsupported digits).

**Additional Metrics.** We introduce three metrics to understand if there are patterns in the mode.

**Mode Probability** is the maximum probability of $\Pi$ averaged across all examples in the dataset, $\mathrm{AVG}_{\text{dataset}} \max \Pi$.

**Mode Stability** is the rate at which the most likely token is preserved between base and chat versions of a model averaged across all examples in the dataset, $\mathrm{AVG}_{\text{dataset}}(\mathbb{1}(\arg\max \Pi_{\text{base}} = \arg\max \Pi_{\text{chat}}))$.

**Mode Frequency** examines whether there are biases for particular tokens (numbers). This metric measures the frequency of the mode averaged across distributions. Because different distributions cover different digits, we group the distributions by underlying problem range and inclusivity: E.g., [2, 5) and [132, 135) are grouped. This is equivalent to: (1) setting the temperature to 0 (greedy sampling), (2) averaging over the groups, keeping the digits dimension, (3) taking the maximum for each group, and (4) averaging across all maximums, $\mathrm{AVG}(\max_{\text{digits}} \mathrm{AVG}_{\text{distributions, keepdim}}(\Pi^{\tau=0}))$.

## 3.5 Experiment: Probabilities

This experiment tests whether the models' output probabilities reflect values in their context.[2] The following is a representative example of the dataset:

> (4) From **17** *red* marbles and **99** *blue* marbles, Billy reached blindly into the bag and grabbed a marble with the color $[red/blue]$

We use five templates, three numeric scales with 100 configurations each, and 110 pairs of options (e.g., red/blue, or orange/purple). The numeric scales are (1) all numbers under ten, (2) ten numbers sampled under 100, and (3) ten numbers sampled under 1000. We these settings bias model behavior. Some factors, such as the numeric context, should influence the model's behavior, while others, such as the ordering of the options, should not.

---

[2]`Probabilities` tests the basic rules of probabilities. As models improve, tests that capture a wider range of probabilities rules would be exciting: Bayes theorem, De Morgan's Laws, Event Independence, etc.

To simplify the analysis of this large number of configurations, we categorize the model behavior into one of six potential reference points based on WD. In practice, we find that models typically follow one of these six behaviors, making this a useful summary of performance.

**Baselines and Reference Points**    First, we create `Random` baselines to contextualize the metric values. Each $\text{Random}_\tau$ baseline: For every row in the dataset, the random baseline produces a $\Pi$ by uniformly sampling two random numbers from 0 to 1 which are then transformed into a distribution parameterized by a temperature $\tau$ using the softmax function. Second, we observe several behaviors in the results; we create a baseline for each of these behaviors (listed below).

0. `Null`: The PM is too low. This can happen for smaller models like `Mistral-7B-v0.1`.
1. `Calibrated`: Sets $\Pi = P$. This is the best case.
2, 3. `Pick Higher`: Sets all probability mass on the $\arg\max \Pi$. This baseline is calibrated directionally: The option with higher probability is correctly preferred but not proportionally. It could alternatively be viewed as mode-collapse. Instead of placing all probability mass on the $\arg\max \Pi$, `Pick Higher`$_{p=q}$ places $q$ (say, 0.7) on $\arg\max$ and $1 - q$ on $\arg\min$ (0.3). In Appendix Table 6, we show performance across a wide range of $p$. (`Pick Lower` analogously places all probability mass on the $\arg\min \Pi$, this rarely occurs.)
4, 5. `Pick First`: Sets all probability mass on the first option, ignoring numeric information. `Pick Second`: analogously places all probability mass on the second option.

## 4    RESULTS

### 4.1    Distributions

There are a few patterns across all models. Table 1 (left) reports the primary metrics. Chat (instruction-tuned) models are less calibrated and have a lower RE than their base counterparts. The base models are more calibrated than the Random baselines. All tested `gpt-*` models have similar calibration scores to the open-source chat models.

We find evidence that the reduced performance of chat models results from their tendency to over-allocate probability on a subset of valid tokens. Table 1 (right) provides additional metrics that suggest mode-collapse. For chat models, the most likely token receives, on average, $66\%$ of the probability mass (up from $28\%$ for base models, beyond the empirical ideal $23\%$, on this dataset). The most likely token remains the same for $48\%$ instances across base and chat models, suggesting that the mode is often preserved.

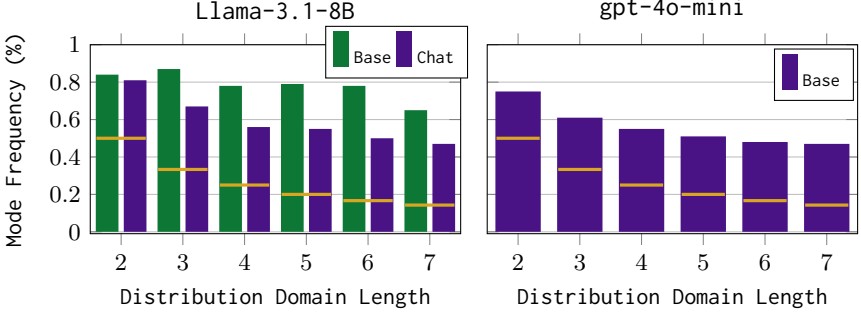

Figure 2: **Models Over-represent Some Numbers**; the modes are heavily over-represented. Each bar is the mode frequency, i.e., how often the top-chosen token is chosen averaged over distributions. The gold lines mark the expected rate for a calibrated model.

Figure 2 reports the mode frequency for two models. If all probability mass were on valid tokens, the minimum mode frequency would be the same as the rate of the uniform distribution; we see much higher rates, ranging from $+29$ to $+50\%$. The implication is that models are also uncalibrated at an outcome level with systematic biases (preferences) for certain numbers. To give an example,

| | PM↑ | | RE↕ | | WD↓ | | Mode Prob. | | Mode |
| --- | --- | --- | --- | --- | --- | --- | --- | --- | --- |
| | Base | Chat | Base | Chat | Base | Chat | Base | Chat | Stable |
| Idealized | 1.00 | | 1.00 | | 0.00 | | 0.23 | 0.23 | - |
| Random$_{\tau=0.1}$ | 1.00 | | 0.30 | | 0.78 | | - | - | - |
| Random$_{\tau=1.0}$ | 1.00 | | 1.60 | | 0.38 | | - | - | - |
| Meta-Llama-3.1-8B | 0.81 | 0.79 | 1.16 | 0.68 | 0.17 | 0.44 | 0.25 | 0.60 | 0.48 |
| Mistral-7B-v0.1 | 0.91 | 0.94 | 1.12 | 1.04 | **0.13** | 0.21 | 0.26 | 0.33 | 0.58 |
| Mistral-7B-v0.3 | 0.90 | **1.00** | 1.12 | 0.52 | **0.14** | 0.56 | 0.26 | 0.65 | 0.35 |
| Mixtral-8x7B-v0.1 | 0.96 | **1.00** | **1.02** | 0.26 | 0.16 | 0.72 | 0.32 | 0.82 | 0.52 |
| Yi-1.5-34B | 0.95 | 0.92 | **1.03** | 0.40 | 0.22 | 0.62 | 0.37 | 0.75 | 0.53 |
| Yi-1.5-9B | 0.90 | 0.95 | 1.09 | 0.47 | 0.18 | 0.63 | 0.30 | 0.75 | 0.49 |
| gemma-2-27b | 0.96 | **1.00** | 1.05 | 0.48 | **0.13** | 0.59 | 0.29 | 0.68 | 0.46 |
| gemma-2-9b | 0.87 | **1.00** | 1.16 | 0.47 | **0.14** | 0.60 | 0.22 | 0.66 | 0.40 |
| AVG$_{OpenSource}$ | 0.91 | 0.96 | 1.09 | 0.57 | 0.16 | 0.58 | 0.28 | 0.66 | 0.48 |
| gpt-3.5 | | **0.99** | | **0.70** | | **0.42** | - | 0.53 | - |
| gpt-4-turbo | | **1.00** | | 0.32 | | 0.69 | - | 0.79 | - |
| gpt-4o | | 0.95 | | 0.59 | | 0.49 | - | 0.58 | - |
| gpt-4o-mini | | 0.73 | | 0.42 | | 0.58 | - | 0.73 | - |
| AVG$_{Proprietary}$ | | 0.92 | | 0.51 | | 0.54 | - | 0.66 | - |

Table 1: **Distributions Results.** Metrics described in Sections 3.2 and 3.4. **Left: (1)** Probability Mass (PM) scores are high for base and chat models, but calibration scores (WD) are lower (better) for base models than chat models. **(2)** The relative entropy decreased significantly with instruction-tuning to $\sim 0.5$ of the ideal value, with **(3)** the entropy of gpt-* models being similarily low. **Right: (1)** Averaged, the probability mass on the top-token for chat models is $+43\%$ above the calibrated ideal, explaining how the relative entropy increased. **(2)** The top-tokens remain the same across base and chat models for 48% of instances. Together, these results suggest a form of mode-collapse.

Appendix Figure 9 shows that for gemma-2-27b 3 is the mode for the distribution [0, 7) in $92\%$ of instances.[3] Appendix Figures 7 to 11 presents the results for all models.

> **Distributions: Systematic Behaviors**
>
> (1) Chat models place a majority of probability mass on a subset of valid tokens;
> (2) The mode is often stable between base to chat models at the instance level.
> (3) We see poor calibration at the instance level and disproportionate outcomes.

## 4.2 Probabilities

**High-level results.** All models perform worse than some simple baselines. Random$_{\tau=1.0}$ performs better than all models. Pick higher$_{p=0.7}$, which allocates 0.7 probability mass to the option associated with the higher value, further outperforms all models. Still, we see interesting performance differences across models.

There are two key parallels between the results in this set of experiments and those in the last section: **(1)** Instruction tuning decreases the entropy of model output probabilities. The decrease in entropy is often due to a large increase of probability on the token with the corresponding higher value (replicating Pick Higher). Unlike the previous experiment, this tends to improve the calibration score overall. Table 2 displays high-level results. **(2)** Model behavior is systematic (and uncalibrated). Understanding (2) requires disaggregating the results and examining model behavior across different colors and color orderings.

**Further Analysis.** To make headway, we characterize the results of two models (Llama-3.1-8B and gpt-4o-mini) in one setting in Figure 3. The full spread of results is available in Appendix C.2. Here, our aim is a high-level understanding of which behaviors best characterize these models. The

---

[3]By looking at Appendix Figure 11, for gemma-2-27b (chat), we can see that the average probability on 3 for [0, 7) is 0.75, also far beyond expectation.

| | PM$_\uparrow$ | | RE$_\updownarrow$ | | WD$_\downarrow$ | |
| --- | --- | --- | --- | --- | --- | --- |
| | Base | Chat | Base | Chat | Base | Chat |
| Idealized | 1.00 | | 1.00 | | 0.00 | |
| Random$_{\tau=0.01}$ | 1.00 | | 0.06 | | 0.69 | |
| Random$_{\tau=0.1}$ | 1.00 | | 0.50 | | 0.55 | |
| pick higher$_{p=0.7}$ | 1.00 | | 1.07 | | 0.15 | |
| pick higher$_{p=1.0}$ | 1.00 | | 0.00 | | 0.47 | |
| Llama-3.1-8B | 0.38 | 0.80 | 1.08 | 0.90 | 0.54 | **0.41** |
| Mistral-7B-v0.1 | 0.30 | 0.54 | 1.10 | 0.74 | 0.59 | 0.50 |
| Mistral-7B-v0.3 | 0.33 | 0.76 | 1.09 | 0.57 | 0.57 | 0.51 |
| Mixtral-8x7B-v0.1 | 0.36 | **0.99** | 1.06 | 0.18 | 0.55 | 0.54 |
| Yi-1.5-9B | 0.30 | **0.99** | 1.05 | 0.33 | 0.59 | 0.52 |
| Yi-1.5-34B | 0.41 | 0.61 | **1.01** | 0.45 | 0.53 | 0.56 |
| gemma-2-9b | 0.39 | **0.99** | 1.14 | 0.49 | 0.53 | 0.54 |
| gemma-2-27b | 0.54 | **1.00** | 1.06 | 0.22 | 0.46 | 0.43 |
| AVG$_{OpenSource}$ | 0.38 | 0.84 | 1.07 | 0.49 | 0.55 | 0.50 |
| gpt-3.5 | | **1.00** | | **0.58** | | **0.32** |
| gpt-4-turbo | | **1.00** | | 0.20 | | 0.45 |
| gpt-4o | | **1.00** | | 0.25 | | 0.42 |
| gpt-4o-mini | | **1.00** | | 0.35 | | 0.42 |
| AVG$_{Proprietary}$ | | 1.00 | | 0.35 | | 0.40 |

Table 2: **Probabilities Results. Takeaways: (1)** Entropy for base models relative to the entropy of a calibrated response is too high (RE above 1.0), whereas for chat models, the entropy is too low (RE below 1.0). However, **(2)** the chat models tend to place more of the probability mass (PM) on valid tokens and **(3)** overall the calibration scores are lower (better).

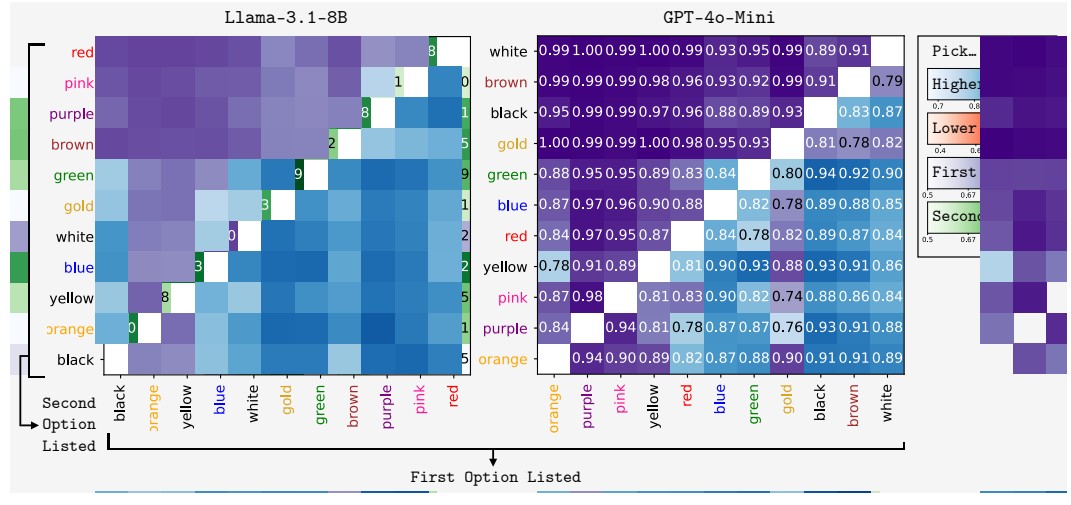

Figure 3: **Systematic Patterns in Model Behavior.** The top-left cell of the left heatmap corresponds to instances where black is listed first and red is listed second across 100 instances. The heatmap color intensity (and numbers) are determined by which behavior the output probabilities most align with as indicated by the legend. For example, 0.97 in the first cell means that 97% of the probability mass is on the second option across those 100 instances. Both models exhibit uncalibrated, biased behavior. Details are discussed in Section 4.2. Representative examples from cells are in Figure 5.

heatmaps in Figure 3 are representative examples; the overall signatures differ mildly across templates and numeric scales (10s, 100s, etc.) and more so across models. For Llama-3.1-8B in Figure 3, the model overly favors the second listed color. More broadly, Llama-3.1-8B favors blue when it is listed second, and avoids white/black/gold. The asymmetry across the diagonal for gpt-4o-mini in

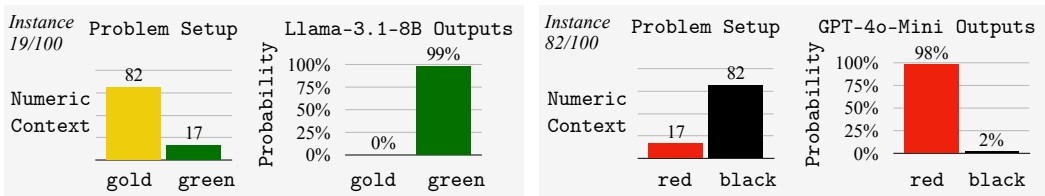

Figure 4: **Representative Examples of Model Behaviors.** These instances are drawn from cells in Figure 3. Left, this is an example of `Pick Second` from gold x green cell for `Llama-3.1-8B`. Right, this is an example of `Pick First` from red x black cell in `gpt-4o-mini`.

Figure 3 shows that model behavior can be dependent on word ordering and that there is a hierarchy among the colors.

The asymmetry across the diagonal for `gpt-4o-mini` in Figure 3 shows that the colors are organized hierarchically. This hierarchy occurs in other models/settings, Appendix C.2. We can read the hierarchy from left to right: (Approximately) each color when listed before a color to its right that first color is `Pick-ed First`, whereas when the order is flipped then there is `Pick Higher`. In this particular example, orange/pink/purple "dominate" most other colors and white is "dominated" by all other colors. This step-wise behavior was neither required nor expected.

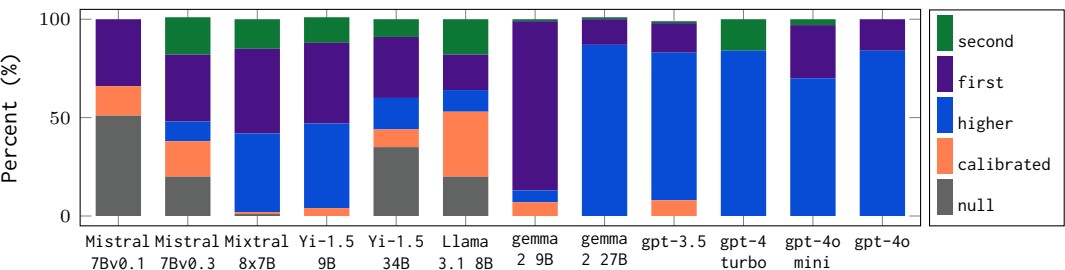

Figure 5: **Systematic Patterns in Model Behavior.** Each bar shows the rates of the different behaviors models exhibit averaged across templates, color pairs, and numeric scales.

Figure 5 shows what behaviors characterize each model: `gpt-*` models (and `gemma-2-27b`) tend to `Pick Higher` whereas others also `Pick First` and `Second`. Smaller models sometimes under-produce probability mass `null`, and there is little calibrated behavior.

> **Probabilities: Systematic Behaviors**
>
> (1) Models often form color hierarchies where the model behavior depends on the word ordering (listed first or second) and the relative rankings of the words in the hierarchy.
> (2) Model behavior closely aligns with heuristic references, often either `Pick First` or `Higher`. In many cases, the heuristic depends on the color ordering.

## 5 HUMAN STUDY

Our experiments so far have demonstrated that these models are generally poorly calibrated. In this section, we try to provide greater context for these results by directly comparing the calibration of proprietary models against humans. We do so using a variation of the storied game, Matching Pennies. The only foolproof approach to winning this game is to behave as randomly as possible. Results are presented in Table 3. The two main takeaways are: 1) Models also produce non-random sequences in this setup, further demonstrating their lack of calibration; 2) When prompted, the model's performance improves and is on par with human performance.

## 5.1 EXPERIMENTAL DESIGN

This experiment is based on a simple heads/tails game. Our motivation for including this experiment is to allow a human baseline in a setting where random behavior is optimal. This section makes no distinction between calibrated probabilities and unbiased outputs because we don't have access to this information for humans. However, to "perform" well, or appear calibrated, requires either the model/person to (1) produce calibrated probabilities which when sampled from will yield random outputs, (2) use an internal random process to produce uncalibrated probabilities that yield random outputs, or (3) have a systematic process for producing outputs that *appear* random enough. (Previous work would suggest that humans are limited in this regard and will likely not be random (and therefore not calibrated).) This leaves us with a one-sided problem. If the humans/models were proficient at producing random outcomes we would have to adjudicate between the above possibilities.

We focus on a variation of the game similar to Eliaz & Rubinstein (2011). Each round, Player 1 aims to match her chosen coin face with Player 2's choice, getting a +1 reward if their coin faces match, and -1 otherwise. In a single-round game, the Nash Equilibrium is to select a coin face randomly. We set there to be 100 rounds of play. To focus on calibration, we frame the tested model as the "misleader." The misleader is prompted to submit answers, and the "guesser" is set to predict the misleader's answers. The misleader submits $m_1, m_2, \ldots m_{100}$ answers, and the guesser bases each of its $g_i$ answers causally based upon all previous misleader's answers, $\{m_j \mid 0 < j < i\}$. We use a n-gram[4] based strategy as our guesser.

**Models**   We focus on `gpt-*` models, sampling temperature $\tau = 1$, and compare them to human behavior. Human behavior is drawn from a sample of 44 games of 100 rounds from $n = 10$ different volunteers with technical backgrounds across up to $t < 10$ different trials.

**Additional Metrics**   Whereas previous metrics rely on model probabilities, this experiment assumes access only to outputs, meaning we cannot use the metrics used in previous sections.

**Player Win Rate (PWR)** is the rate the misleader defeats the guesser, averaging over 100 games each for up to 100 rounds. The higher this rate, the more random the outputs of the misleader, and thus the more calibrated it is at an outcome level.

**Randomness Testing (RNG)**. The Wald–Wolfowitz[5] runs test checks if the elements of a sequence are mutually independent, and can often rule out human-generated sentences as non-random. Using this test at a significance level of 0.05, we report the proportion of samples that cannot statistically be rejected as non-random (note this does not guarantee them to be random) and label this value as **RNG**. For reference, as shown in Table 3 when we sampled random sequences using `python`, $98\%$ of the sequences could not be rejected; only $7\%$ of human-generated samples could not be rejected.

## 5.2 RESULTS

When a model is instructed to be random (denoted by "+instr") Player Win Rate (PWR) improves. For `gpt-4o`, PWR increases by $+30\%$. Explicit instruction following ability can thus improve calibration at the outcome level. Notably, when prompted, model performance is similar to human performance.

Table 3 reports the proportion of samples (where each sample is 100 coin flips), at $p = 0.05$, the result could not be considered non-random: Higher RNG values suggest the model is more random. More model- than human-generated samples were considered random, $7\%$ vs $19 - 43\%$.

> **Humans and Proprietary (OpenAI) Models Are Both Non-Random (*Un*calibrated)**
>
> (1) Even top-performing proprietary models are unable to produce random behavior
> (2) Human behavior is also non-random. Even if models are currently similar to humans in this regard, we argue that future work endowing models with the ability to produce calibrated outputs would improve human interactions with models.

---

[4]Motivated by the strategy here: https://www.expunctis.com/2019/03/07/Not-so-random.html.
[5]Results with Bartel's test perfectly correlate with the runs test so we only report the runs test.

| | PWR$_\uparrow$ | | RNG$_\uparrow$ | |
|---|---|---|---|---|
| | base | +instr. | base | +instr. |
| Random | 0.50 | | 0.98 | |
| Human | 0.33 | | 0.07 | |
| gpt-3.5 | 0.26 | **0.40** | 0.31 | **0.43** |
| gpt-4-turbo | 0.02 | 0.33 | 0.00 | 0.25 |
| gpt-4o | 0.06 | 0.36 | 0.08 | 0.19 |

Table 3: **Human Study Results: Player win rates (PWR) and statistical tests for randomness (RNG).** Higher values for both metrics suggest that models are more able to produce random (and thus more calibrated) outputs. **Takeaways: (1)** `gpt-3.5` stands out as the most random model; **(2)** Models pass a greater number of the statistical tests for randomness than humans, though are still far from random; **(3)** Model sequences are predictable at rates similar to humans; **(4)** Adding instructions to be random leads to more random outputs.

## 6 DISCUSSION

**Why Aren't LLMs Already Calibrated?**   Should the negative log-likelihood loss endow a pre-trained language model with calibration? If there were $n \to \infty$ examples of Heads and Tails being flipped in the data and the true distribution was reflective of some $p_{\text{heads}}$, then the lowest loss would be achieved via a perfectly calibrated model. Thus, the answer is empirical: What does the training data look like? Finding the disparate scenarios that call for random outcomes may be difficult. Non-calibrated outputs frequently occurring in such scenarios might harm numeric calibration. For example, patterns derived from people preferring the number 7 (out of 10) or 42 (out of 100) or Heads (over Tails) or answering with round numbers ending in 0 or 5 might reduce calibration. Possibly, some models are capable of producing calibrated results; the models may "know" the calibrated output but instead match non-random tendencies, akin to pragmatics.

**Should Models Be Calibrated?**   Our work highlights how models that perform remarkably in other types of evaluations fall short in scenarios that call for calibrated outputs. Some may have expected our results given that these models are trained on data generated by people who are also widely biased. Others might have expected that with the range of data seen in training (data in the wild, code outputs, endemic instances of distributions) models would learn to calibrate. However, it seems curious that models are frequently able to correctly generate text describing what the proper calibrated distribution should be for a given scenario, but fail to represent that distribution internally when prompted to simulate the same scenario.

**Can We Calibrate Models with FineTuning or using Tools?**   We expect that it is possible to fine-tune models to improve calibration. Already we see some improvements with generic alignment procedures, though these improvements often come at a tradeoff of an (overly large) reduction in model entropy. Mitigating this mode collapse behavior is a compelling area for future work. Using tools (Schick et al., 2024) is another practical solution for simple scenarios like those we present, but may not obviously generalize to more complex scenarios.

## 7 CONCLUSION

All models we test are not only *not* calibrated, but behave in different systematically uncalibrated ways. A common theme is that top-performing models (e.g., gemma-2-27b and `gpt-*` models) `Pick Higher`, allocating most of the probability mass upon single output token. This is an example of mode collapse. Though mode-collapse is commonly studied in computer vision and image generation (Thanh-Tung & Tran, 2020; Srivastava et al., 2017) *inter alia*, O'Mahony et al. (2024) show how RLHF reduces diversity in model outputs, leading to this problem in text generation. Our results are similar: Instruction-tuned models produce valid outputs, but the probability distribution is collapsed to a small portion of the valid answers that are equally correct. While we focus on controlled problem settings, these issues leave vulnerabilities. For instance, an adversary could leverage that a model is going to behave in a systematically arbitrary manner.

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

## A  DETAILS

All data and experiments will be released at `https://drive.google.com/drive/folders/1jAcKRqAuv61syDaCtkSXbq3KBYwrzIW_?usp=sharing`. All details required for reproducing our results like problem templates and prompts will also be released there. Appendix C.2 report the prompts (formatted as messages) and problem templates.

We use the models listed in Table 4. For visual acuity, we put the full model codes into this table rather than throughout the paper.

## B  HUMAN STUDY

Figure 6 reports our the implementation for the n-gram model. We find that most n from 1 to 5 do similarly well. In the paper we report a 4-gram model.

## C  MORE RESULTS

### C.1  Distributions

Mode preservation and frequency results are in Figures 7 to 11, extending the results shown in Figure 2 to other models. For every model, models over represent some of the numbers, with an average excess probability of 29 to 50%. Figures 8 to 11 disaggregate these results. Figure 10 vs Figure 11 show that while chat models place far more probability mass on a single token, the outcomes (how often a given token is chosen, the mode) is still heavily biased even for base models, Figure 8 and Figure 9.

### C.2  Probabilities

Table 6 has high level results for the baselines across a wider range of settings.

Figure 16 shows model color preferences (or differences) across all models; the regularity of the results seen is striking. This figure shows that models will consistently be more calibrated for some colors over others; exactly how this translates into model behavior requires more nuance.

Figures 19 to 30 show this nuance with in-depth heatmap performances for all models across all templates (columns) and numeric scales (grids). The legend for these latter plots is shown in Figure 17. This set of results is extremely long. We provide a (Space Station) view of these results in Figure 18 which shows grouped and compressed figures. Only the general trends as shown by the cell colors can be discerned. What we can see is that the primary takeaways from this set of figures is that 1) for a given model, there is a lot of stability across numeric scales (10s, 100s, etc.) (columns per model are similar) and somewhat less stability across different templates/prompts (rows per model can differ). 2) Models tend to have different patterns of results from each other. 3) Pick first, pick higher are most common followed by pick second. Many but not all models have obvious cross-diagonal asymmetries.

***The remainder of this document are figures referenced by the text above.***

| Our Name | HF Model Key (Base) | HF Model Key (Chat) |
|---|---|---|
| Yi-1.5-9B | 01-ai/Yi-1.5-9B | 01-ai/Yi-1.5-9B-Chat |
| Yi-1.5-34B | 01-ai/Yi-1.5-34B | 01-ai/Yi-1.5-34B-Chat |
| Llama-3.1-8B | meta-llama/Llama-3.1-8B | meta-llama/Llama-3.1-8B-Instruct |
| Mistral-7B-v0.1 | mistralai/Mistral-7B-v0.1 | mistralai/Mistral-7B-Instruct-v0.1 |
| Mistral-7B-v0.3 | mistralai/Mistral-7B-v0.3 | mistralai/Mistral-7B-Instruct-v0.3 |
| Mixtral-8x7B-v0.1 | mistralai/Mixtral-8x7B-v0.1 | mistralai/Mixtral-8x7B-Instruct-v0.1 |
| gemma-2-9b | google/gemma-2-9b | google/gemma-2-9b-it |
| gemma-2-27b | google/gemma-2-27b | google/gemma-2-27b-it |
| | | OpenAI Model Key |
| gpt-3.5 | | gpt-3.5-turbo-0125 |
| gpt-4-turbo | | gpt-4-turbo-2024-04-09 |
| gpt-4o | | gpt-4o-2024-08-06 |
| gpt-4o-mini | | gpt-4o-mini-2024-07-18 |

Table 4: **Models**

| | $RE_{\updownarrow}$ | $PM_{\uparrow}$ | $WD_{\downarrow}$ |
|---|---|---|---|
| Idealized | 1.00 | 1.00 | 0.00 |
| $Random_{\tau=0.01}$ | 0.10 | 1.00 | 0.86 |
| $Random_{\tau=0.1}$ | 0.31 | 1.00 | 0.78 |
| $Random_{\tau=1.0}$ | 1.61 | 1.00 | 0.38 |

Table 5: **Baselines on `Distributions` Results.**

| | $PM_{\uparrow}$ | $RE_{\updownarrow}$ | $WD_{\downarrow}$ |
|---|---|---|---|
| Idealized | 1.00 | 1.00 | 0.00 |
| `pick first`$_{p=1.0}$ | 1.00 | 0.00 | 0.71 |
| `pick second`$_{p=1.0}$ | 1.00 | 0.00 | 0.71 |
| `pick higher`$_{p=0.6}$ | 1.00 | 1.17 | 0.16 |
| `pick higher`$_{p=0.65}$ | 1.00 | 1.13 | 0.15 |
| `pick higher`$_{p=0.7}$ | 1.00 | 1.07 | 0.15 |
| `pick higher`$_{p=0.75}$ | 1.00 | 0.98 | 0.18 |
| `pick higher`$_{p=0.8}$ | 1.00 | 0.87 | 0.21 |
| `pick higher`$_{p=0.85}$ | 1.00 | 0.74 | 0.26 |
| `pick higher`$_{p=0.90}$ | 1.00 | 0.46 | 0.36 |
| `pick higher`$_{p=1.0}$ | 1.00 | 0.00 | 0.47 |
| `pick lower`$_{p=1.0}$ | 1.00 | 0.00 | 0.95 |
| $Random_{\tau=0.01}$ | 1.00 | 0.06 | 0.69 |
| $Random_{\tau=0.1}$ | 1.00 | 0.50 | 0.55 |
| $Random_{\tau=1.0}$ | 1.00 | 1.17 | 0.26 |
| $Random_{\tau=2.0}$ | 1.00 | 1.20 | 0.25 |
| $Random_{\tau=10.0}$ | 1.00 | 1.21 | 0.24 |

Table 6: **Baseline `Probabilities` Results.**

```python
CoinFace = str, H: CoinFace = "H", T: CoinFace = "T"
def flip(face: CoinFace) -> CoinFace:
    return {H: T, T: H}[face]
def mdp_1gram(sequence: List[CoinFace]) -> CoinFace:
    return _mdp_ngram(sequence, 1)
def mdp_2gram(sequence: List[CoinFace]) -> CoinFace:
    return _mdp_ngram(sequence, 2)
def mdp_3gram(sequence: List[CoinFace]) -> CoinFace:
    return _mdp_ngram(sequence, 3)
def mdp_4gram(sequence: List[CoinFace]) -> CoinFace:
    return _mdp_ngram(sequence, 4)
def mdp_5gram(sequence: List[CoinFace]) -> CoinFace:
    return _mdp_ngram(sequence, 5)
def _mdp_ngram(sequence: List[CoinFace], ngram_size: int) -> CoinFace:
    """Picks the most common continuation for a given sequence based upon all n_grams
    ↪  of the current size.

    Defaults to H if sequence is empty. Flips previous coinface if the length of the
    ↪ sequence is 1.
    """
    assert ngram_size >= 1

    # Handle short sequences
    if len(sequence) == 0:
        return H
    elif len(sequence) == 1:
        if sequence[0] == H:
            return T
        else:
            return H
    if len(sequence) <= ngram_size:
        return _mdp_ngram(sequence, min(ngram_size - 1, len(sequence) - 1))

    map_ = {H: 0, T: 1}
    sequence_numbers = [map_[face] for face in sequence]
    mdp = np.zeros([2 for _ in range(ngram_size + 1)])
    for i in range(0, len(sequence_numbers) - ngram_size):
        index = tuple(sequence_numbers[i : i + (ngram_size + 1)])
        mdp[index] = mdp[index] + 1

    last = tuple(sequence_numbers[-ngram_size:])
    transitions = mdp[last]
    if transitions[0] > transitions[1]:
        return H
    elif transitions[0] == transitions[1]:
        # NOTE: Alt, we could pick which ever is more common
        # or fall back to a lower ngram or sample.
        return H
    else:
        return T
```

n-gram model implementation

Figure 6: **n-gram model implementation.**

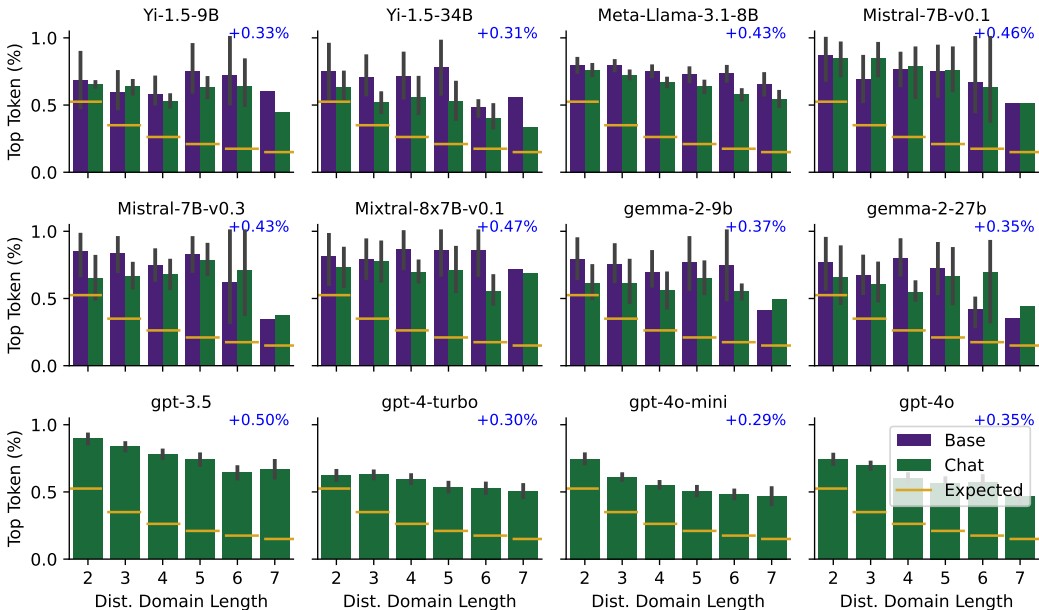

Figure 7: **Models Over-represent Numbers.** Each bar shows how often the top-chosen token is chosen (percent). The gold bars mark the expected rate for a calibrated model; the blue annotation marks the average excess probability on the top-chosen token, ranging from $29 - 50\%$ across models. Takeaways **(1)**: Models over-represent a token (number) over other valid options. Notes: See disaggregated results in Appendix Figure 8, Figure 9, Figure 10, Figure 11. This token is not always the same but the pattern of over-representing a number irrespective of the numeric context holds.

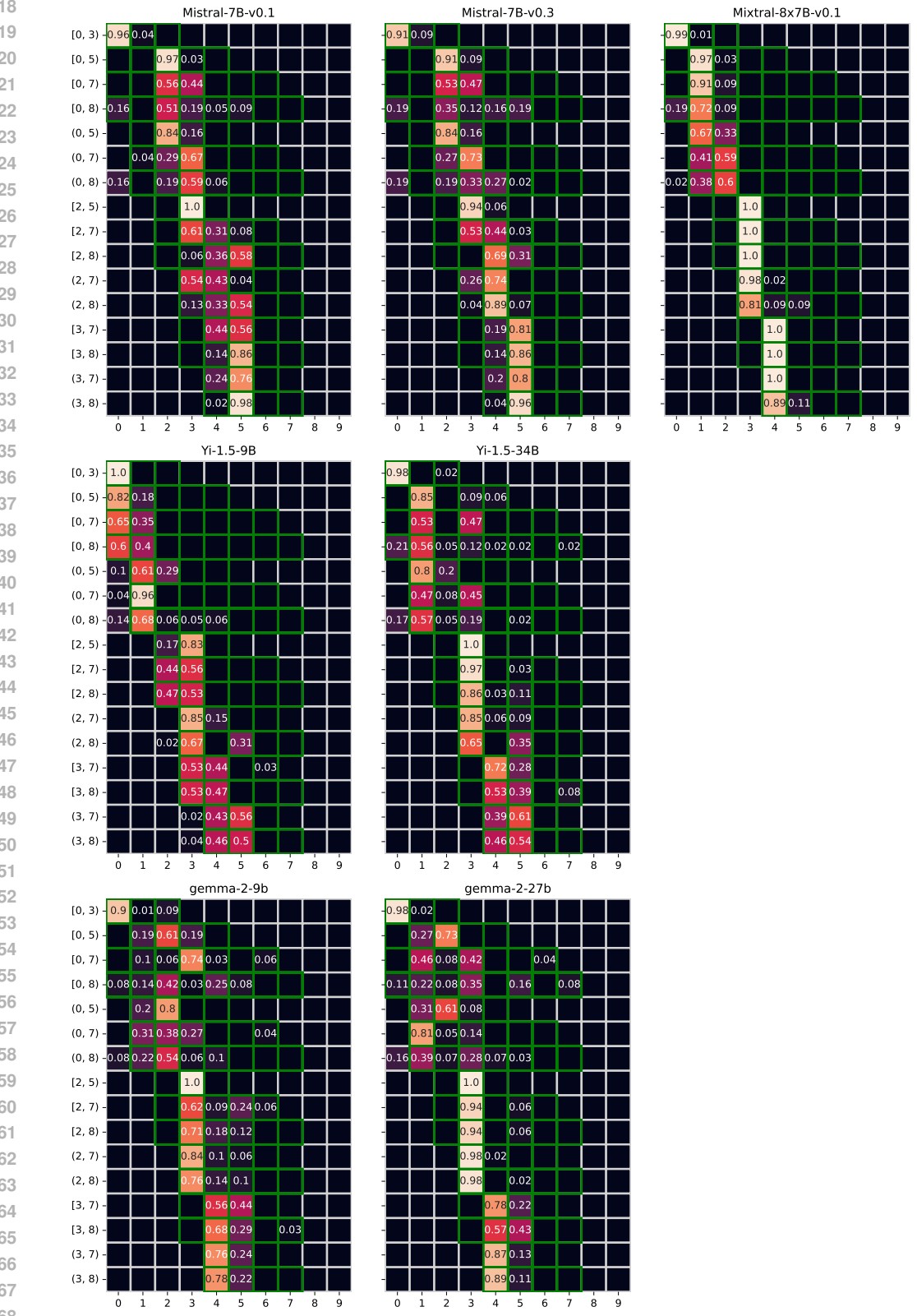

Figure 8: **Top Tokens; Base Models.** This plot is read across rows. Each row contains all distributions that end with the given digits and the denoted inclusivity. For example, the 8th row, marked [2, 5), tracks examples like [12, 15) and [122, 125). The green outlined cells are supported digits. All zero-valued cells are empty for visual acuity. The expected result is uniform results across the supported cells in each row.

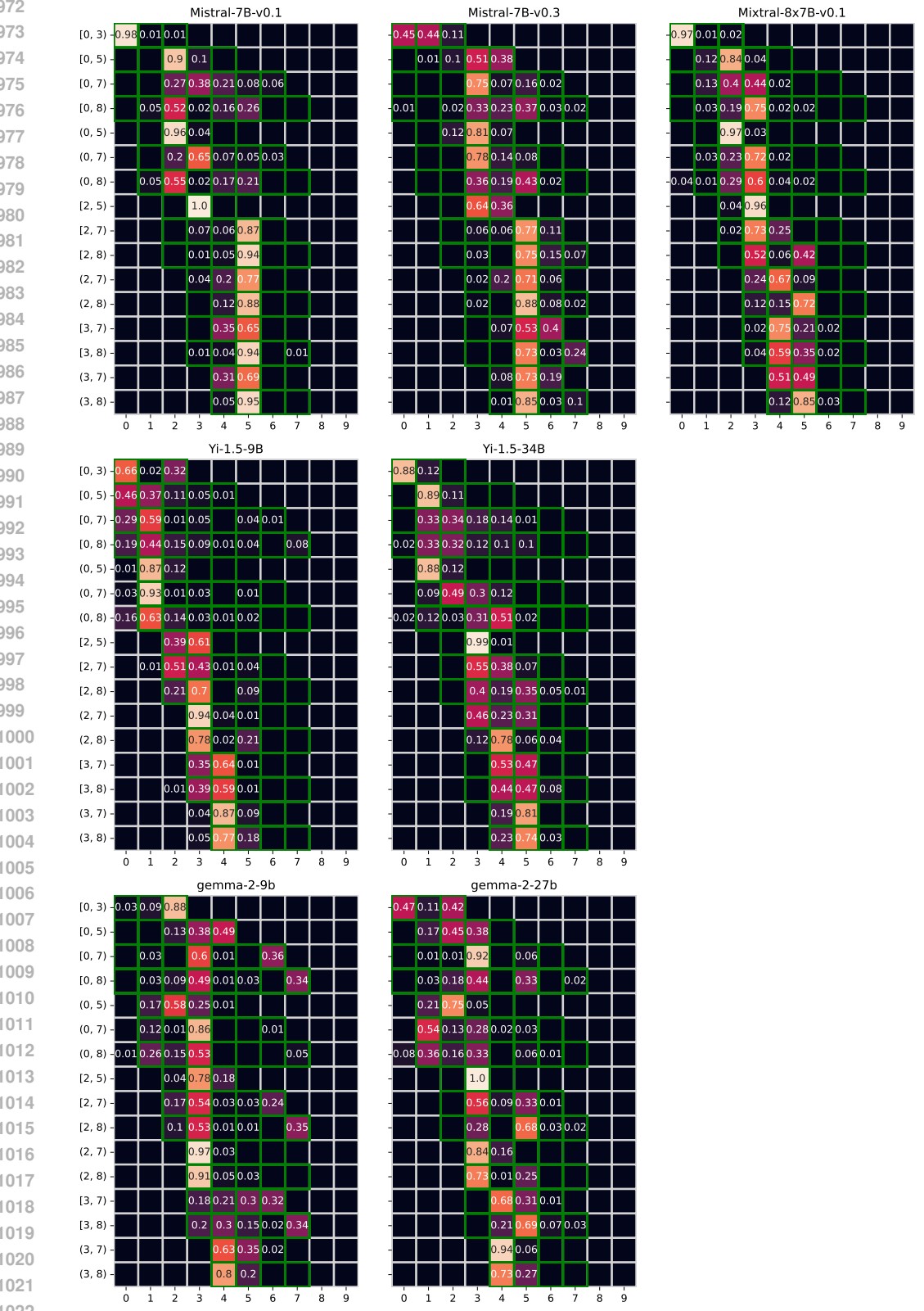

Figure 9: **Top Tokens; Chat Models.** This plot is read across rows. Each row contains all distributions that end with the given digits and the denoted inclusivity. For example, the 8th row, marked [2, 5), tracks examples like [12, 15) and [122, 125). The green outlined cells are supported digits. All zero-valued cells are empty for visual acuity. The expected result is uniform results across the supported cells in each row.

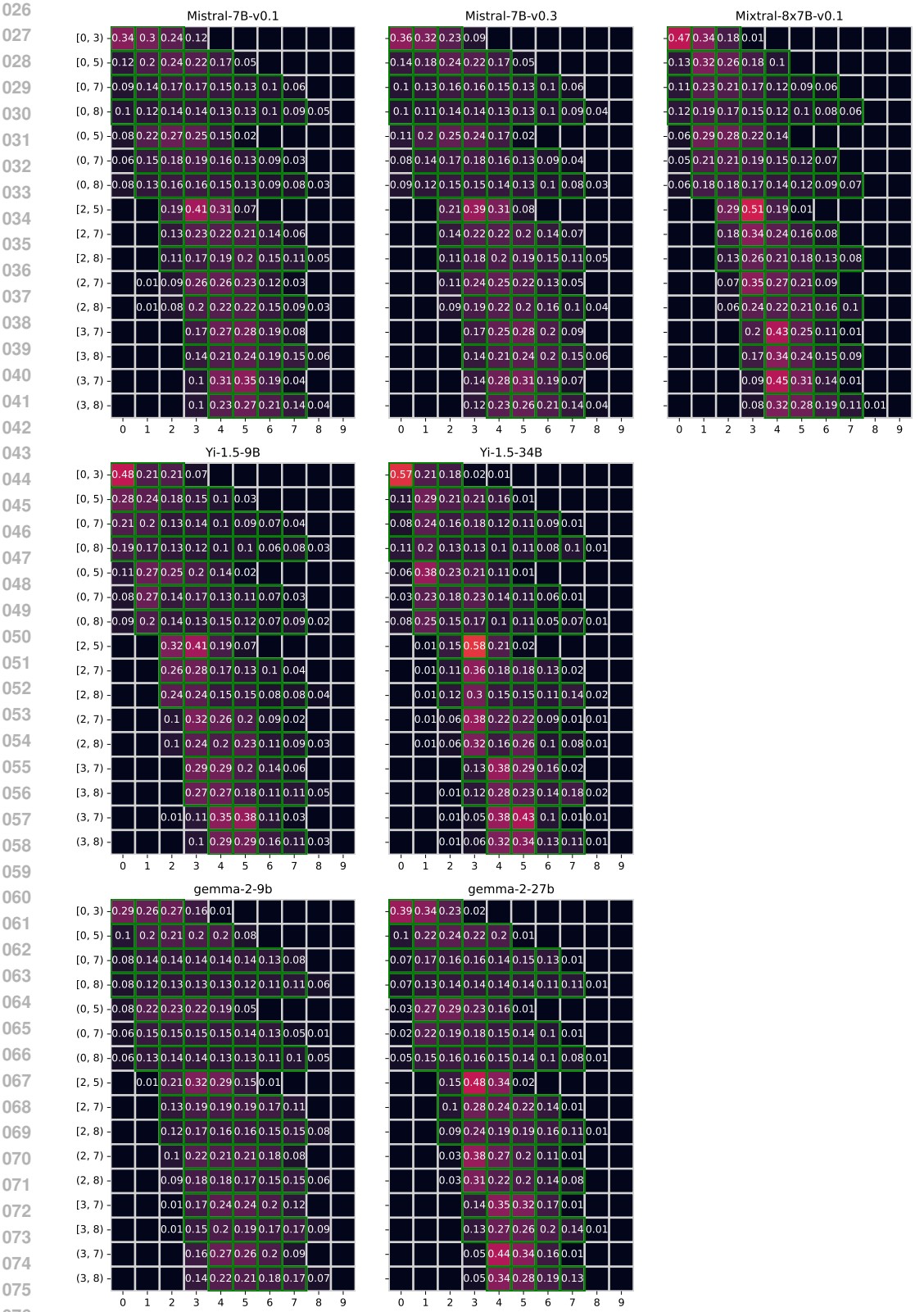

Figure 10: **Probabilities; Base Models.** This plot is read across rows. Each row contains all distributions that end with the given digits and the denoted inclusivity. For example, the 8th row, marked [2, 5], tracks examples like [12, 15) and [122, 125). The green outlined cells are supported digits. All zero-valued cells are empty for visual acuity. The expected result is uniform results across the supported cells in each row.

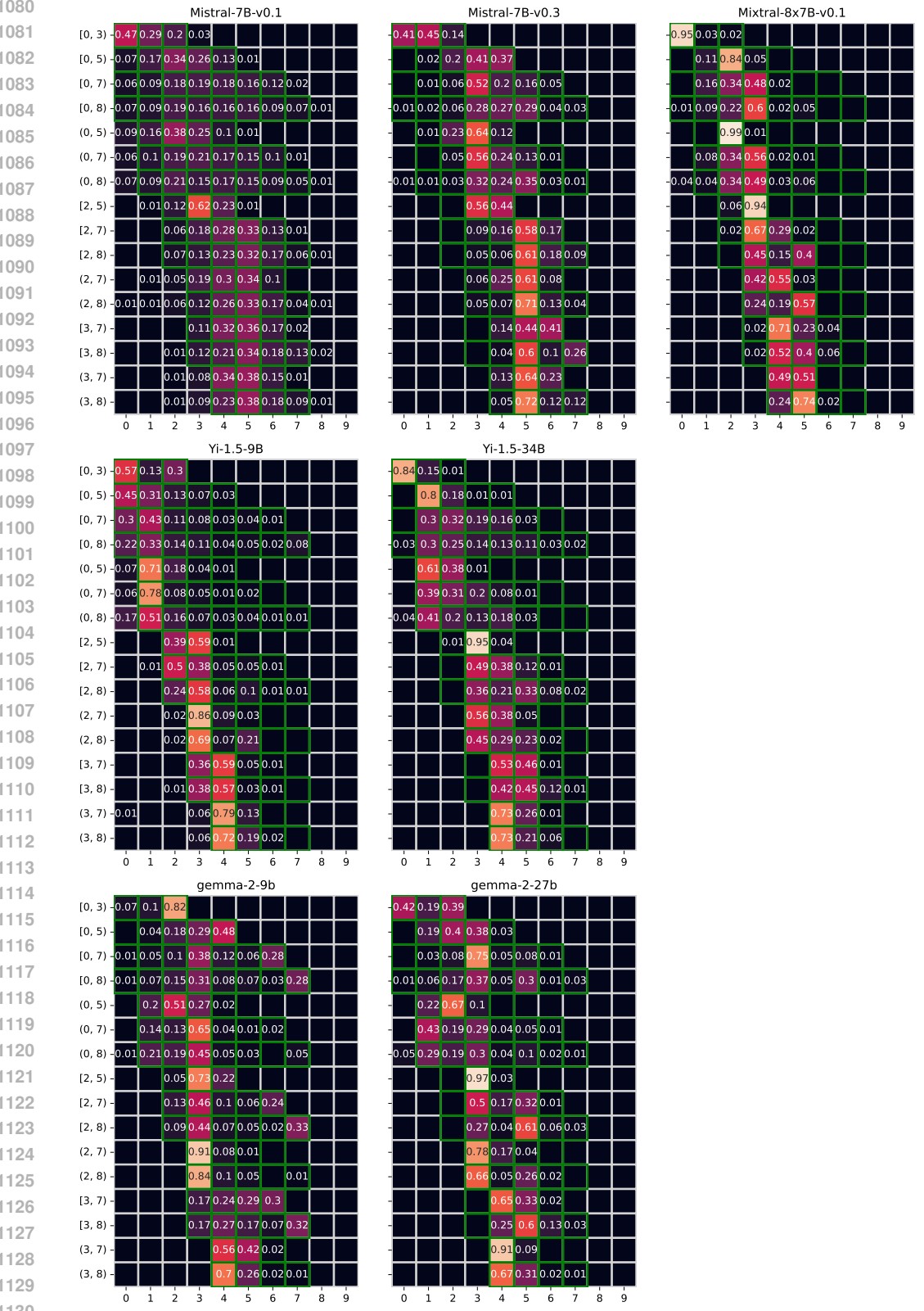

Figure 11: **Probabilities; Chat Models.** This plot is read across rows. Each row contains all distributions that end with the given digits and the denoted inclusivity. For example, the 8th row, marked [2, 5), tracks examples like [12, 15) and [122, 125). The green outlined cells are supported digits. All zero-valued cells are empty for visual acuity. The expected result is uniform results across the supported cells in each row.

```
# prompt 0
[
 {
 "role": "user",
 "content": "You are a perfectly calibrated language model that produces tokens with
     ↪ probabilities that correspond to values in the context and prompt. I will
     ↪ give you a sentence. Finish the sentence by producing only a number and
     ↪ nothing else.",
 },
 {
 "role": "assistant",
 "content": "Excellent. I will continue by producing a single number after your next
     ↪ message.",
 },
 {"role": "user", "content": @text},
]
# prompt 1
[
 {
 "role": "user",
 "content": "You are a master of statistics and math. Produce tokens perfectly
     ↪ calibrated to the numbers in context.  I will give you a sentence. Produce
     ↪ one number to finish the sentence.",
 },
 {"role": "assistant", "content": "Excellent. Ask away!"},
 {"role": "user", "content": @text},
]
# prompt 2
[
 {
 "role": "user",
 "content": "Produce tokens with probability calibrated to the numbers in context.
     ↪ Given the scenario, produce a number.",
 },
 {"role": "assistant", "content": "Excellent. Go ahead!"},
 {"role": "user", "content": @text},
]
```

We use the message format shared by OpenAI and HuggingFace. The problem {text} is injected into the prompt message.

Figure 12: **Prompts For Instruction-Tuned Models; `Distributions`.**

```
# prompt 0
[
 {
 "role": "user",
 "content": "You are a perfectly calibrated language model that produces tokens with
     ↪ probabilities that correspond to values in the context and prompt. I will
     ↪ give you a scenario. Produce only a single word and nothing else. The word
     ↪ will be one of the options in context.",
 },
 {
 "role": "assistant",
 "content": "Excellent. I will continue by producing a single word after your next
     ↪ message.",
 },
 {"role": "user", "content": text},]
# prompt 1
[
 {
 "role": "user",
 "content": "You are a perfectly calibrated language model that produces tokens with
     ↪ probabilities that correspond to values in the context and prompt. I will
     ↪ give you a context where there are some probabilities involved. Produce one
     ↪ token.",
 },
 {
 "role": "assistant",
 "content": "Excellent. I will generate a single token calibrated to the statistics
     ↪ of the context.",
 },
 {"role": "user", "content": text},]
# prompt 2
[
 {
 "role": "user",
 "content": "You are a master of statistics and math. Produce tokens perfectly
     ↪ calibrated to the numbers in context. Produce one token.",
 },
 {"role": "assistant", "content": "Excellent. Ask away!"},
 {"role": "user", "content": text},]
# prompt 3
[
 {
 "role": "user",
 "content": "Produce tokens with probability calibrated to the numbers in context.
     ↪ Produce a single token.",
 },
 {"role": "assistant", "content": "Excellent. Ask away!"},
 {"role": "user", "content": text},]
# prompt 4
[
 {
 "role": "user",
 "content": "Produce tokens with probability calibrated to the statistics implicit
     ↪ and explicit in context. Produce only a single token.",
 },
 {"role": "assistant", "content": "Excellent. Ask away!"},
 {"role": "user", "content": text}, ],
```

We use the message format shared by OpenAI and HuggingFace. The problem {text} is injected into the prompt message.

Figure 13: **Prompts For Instruction-Tuned Models; `Probabilities`.**

```
# template 0
"Picking an integer between {number_1} inclusive and {number_2} exclusive, the
    ↪ integer I got was{prefix}"
# template 1
"Picking an integer between {number_1} and {number_2}, exclusive on both sides, the
    ↪ integer I got was{prefix}"
# template 2
"I sampled an integer from a random variable distributed uniformly between {number_1}
    ↪  inclusive and {number_2} exclusive, getting the value{prefix}"
# template 3
"I sampled an integer from a random variable distributed uniformly between {number_1}
    ↪  and {number_2}, exclusive on both ends, getting the value{prefix}"
```

The variables are injected into the text; {number_1}, {number_2}, {prefix}. {prefix} is always set to empty; early pilot experiments suggest that using an ellipsis leads to a similar pattern of results.

Figure 14: **Templates for `Distributions`.**

```
# template 0
"There were {number_1} {option_1} marbles and {number_2} {option_2} marbles in a bag.
    ↪  Jane randomly picked a marble and saw it was the color"
# template 1
"Billy got to pick one marble from a big urn with many marbles. There were {number_1}
    ↪ {option_1} marbles and {number_2} {option_2} marbles in an urn. The color of
    ↪  the marble Billy randomly picked was"
# template 2
"Amanda had a huge pile of shirts. There were {number_1} {option_1} shirts and {
    ↪ number_2} {option_2} shirts. Without looking, she picked one by chance. The
    ↪ color of the shirt was"
# template 3
"Bill and Rick went to the hardwore store for paint in a hurry. The store had {
    ↪ number_1} shades of {option_1} and {number_2} shades of {option_2}. They didn'
    ↪ t have any time to test out colors so they randomly grabbed a can. The color
    ↪ they grabbed turned out to be"
# template 4
"Kids at soccer practice randomly grabbed pinnies from a bag. There were {number_1} {
    ↪ option_1} pinnies and {number_2} {option_2} pinnies. Tommy's pinny was the
    ↪ color"
```

The variables are injected into the text; {number_1}, {number_2}, {option_1}, {option_2}.

Figure 15: **Templates for `Probabilities`.**

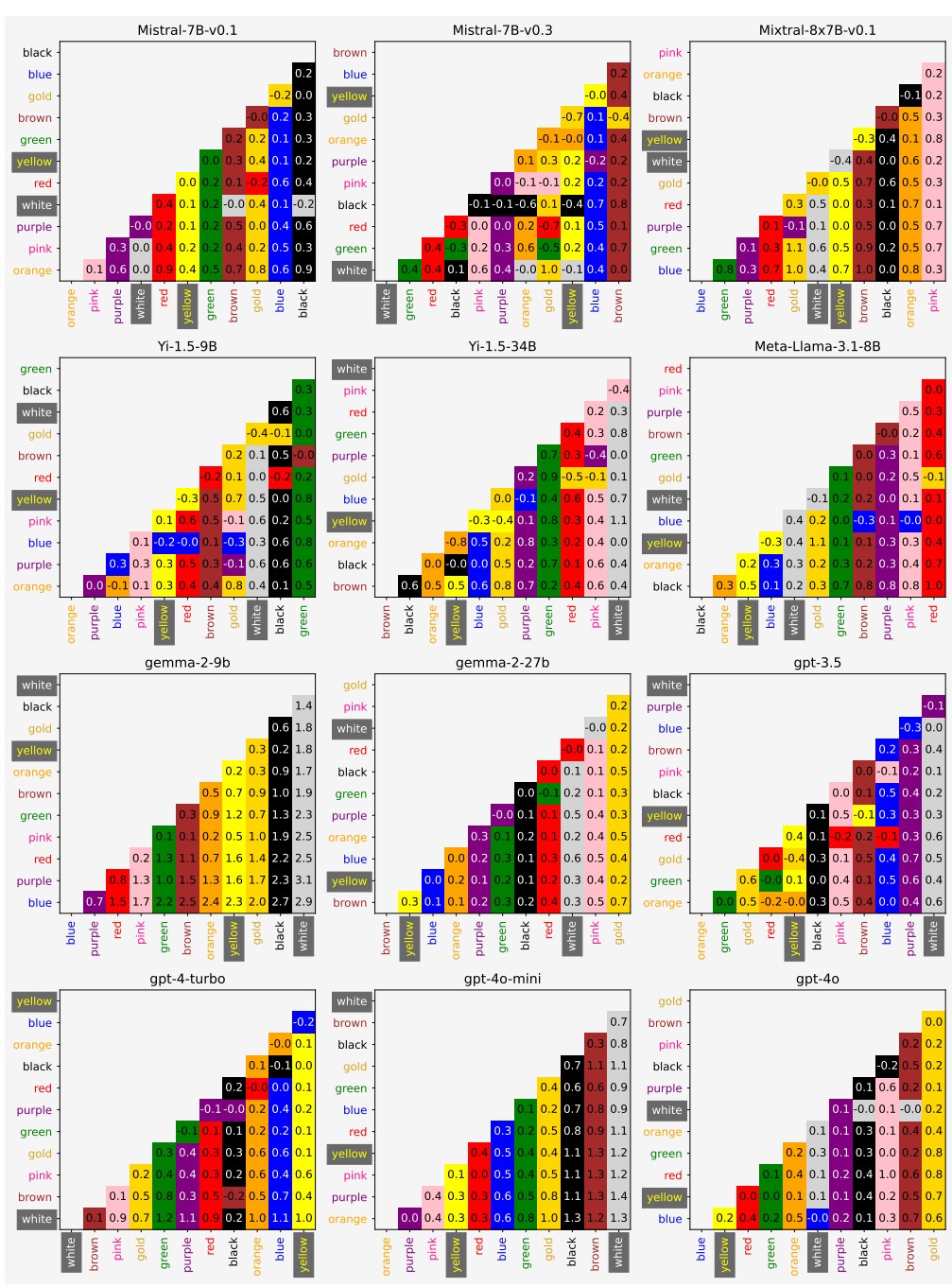

Figure 16: **Model Color Preferences: Differences in model calibration scores when one color is first vs another.** Higher scores indicate that the "winning" color, shown on the x-axis, has a comparatively better fit when listed first versus second. **Takeaways: (1)** Almost all models form hierarchies across colors where certain colors always lead to better fits when listed first versus second. (Note) This does not mean that any given color is picked more than other colors, just that there is an order bias where certain colors lead to more calibrated behaviors. When we further expand out these results like into the heatmaps in the main body of the paper, we can better understand what the model behavior actually looks like. Because the ordering of the colors is different across all models, without much shared structure, we do not have an obvious hypothesis about why these particular color hierarchies arose.

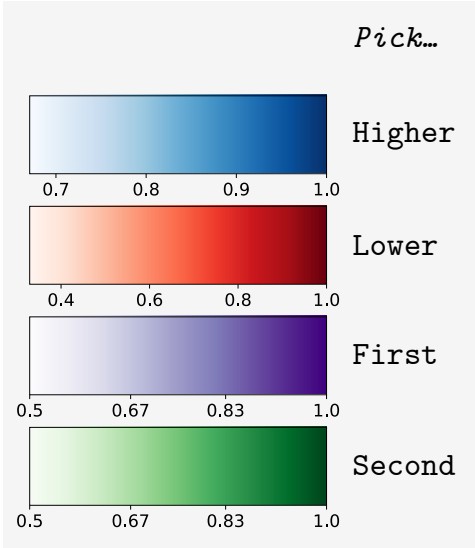

Figure 17: **Legend for Systematic Patterns in Model Behavior Heatmaps.** The cells in the following colors heatmaps are determined by profile most aligned with the model behavior. Especially in some cases, for models like mistral-7b-v0.1, the model behavior is not overly similar to any profile. This is largely because the probability mass is relatively low See the plots in Figures 19 to 30.

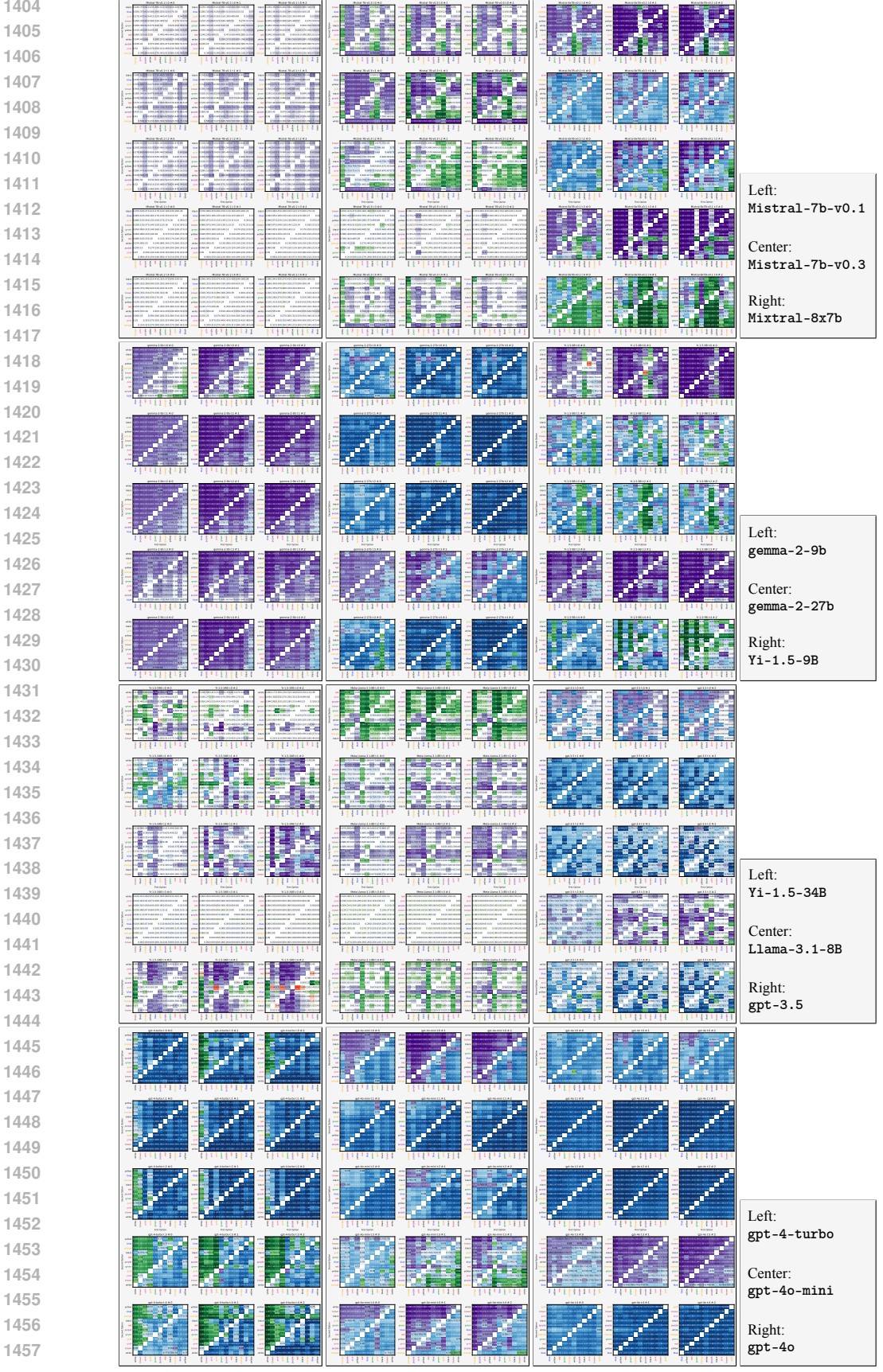

Left:
`Mistral-7b-v0.1`

Center:
`Mistral-7b-v0.3`

Right:
`Mixtral-8x7b`

Left:
`gemma-2-9b`

Center:
`gemma-2-27b`

Right:
`Yi-1.5-9B`

Left:
`Yi-1.5-34B`

Center:
`Llama-3.1-8B`

Right:
`gpt-3.5`

Left:
`gpt-4-turbo`

Center:
`gpt-4o-mini`

Right:
`gpt-4o`

Figure 18: **Space Station View of Model Behaviors.**

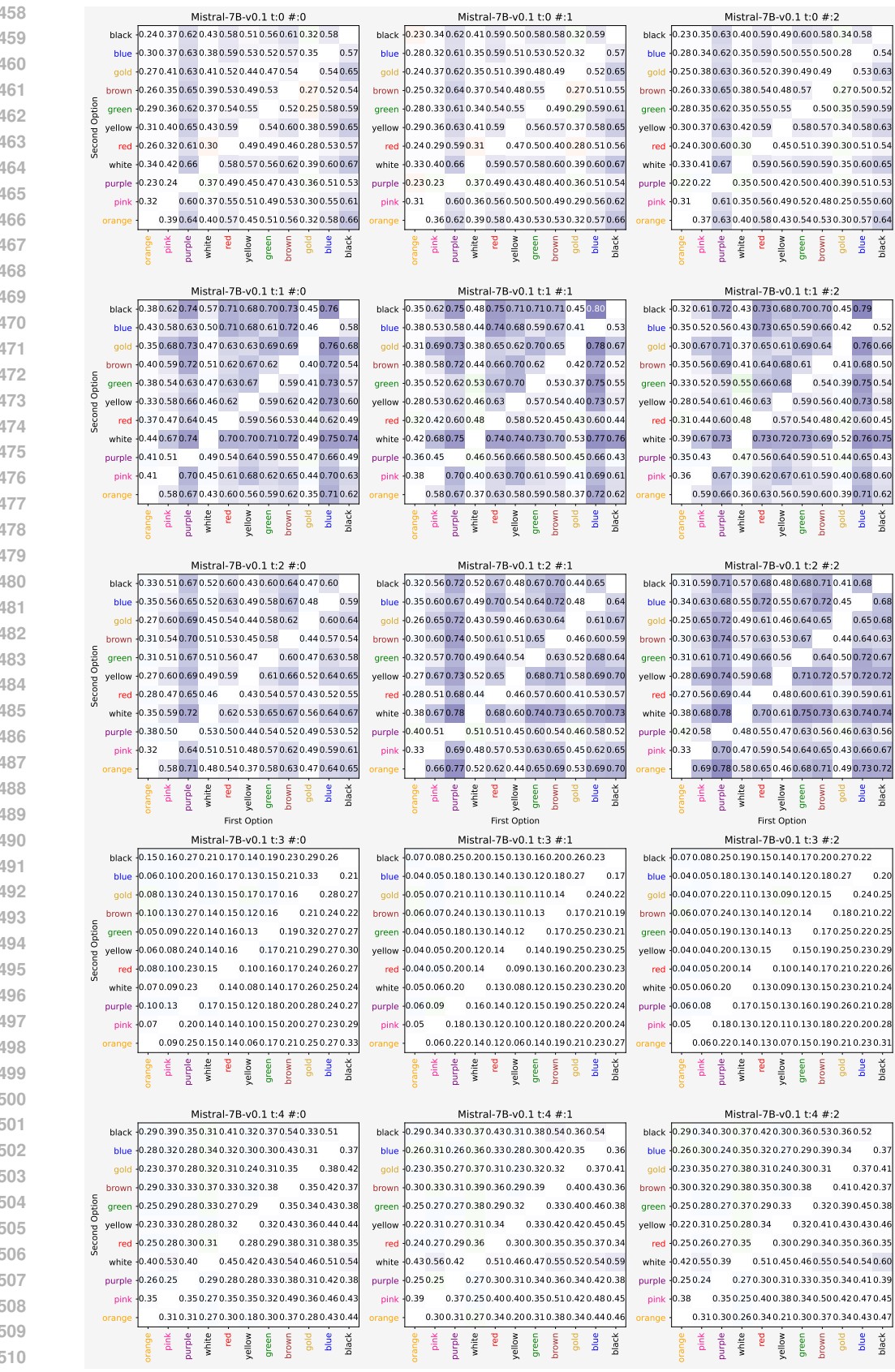

Figure 19: **Model Behavior Profile: mistral-7b-v0.1 (chat).**

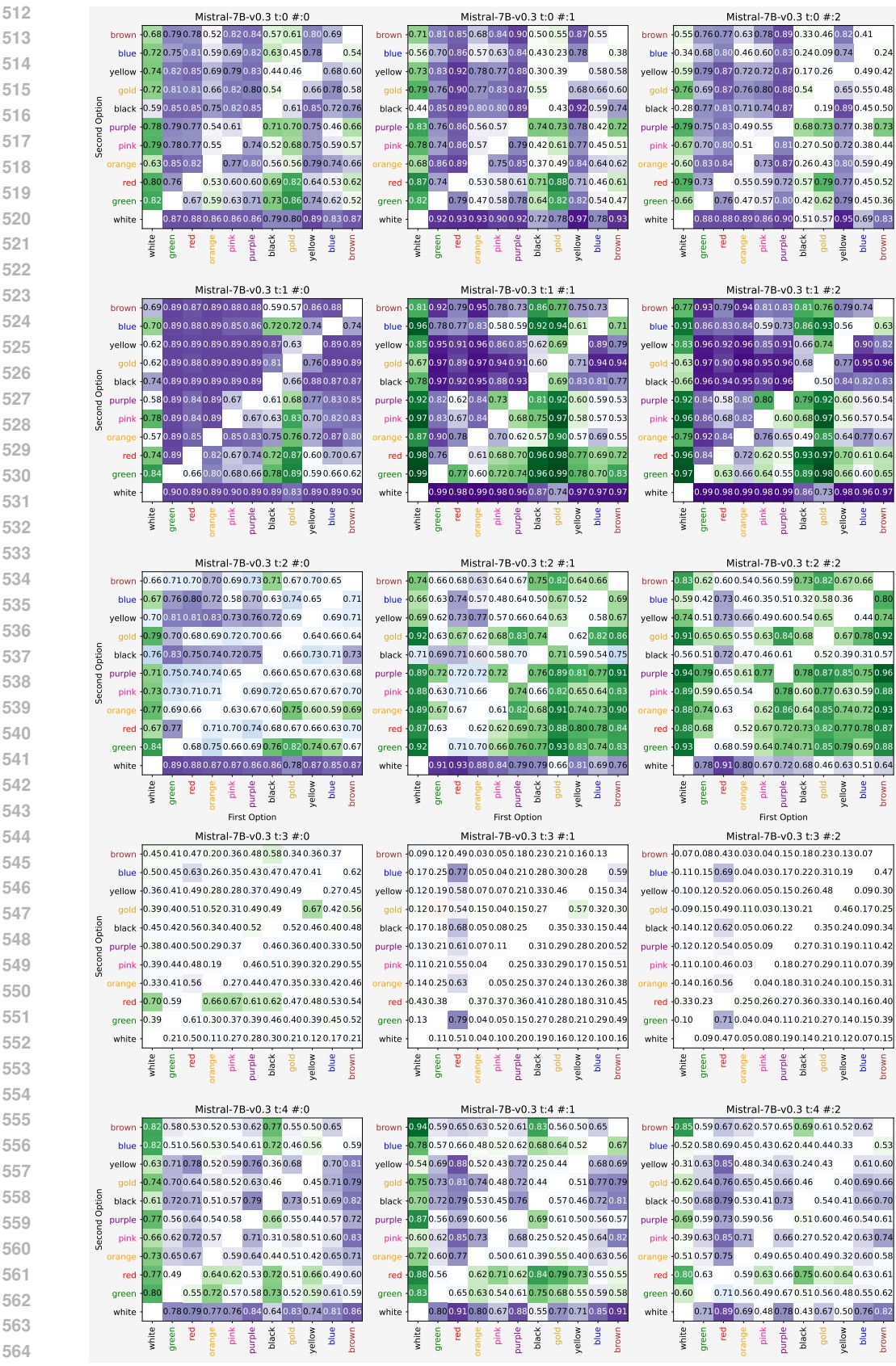

Figure 20: **Model Behavior Profile: mistral-7b-v0.3 (chat).**

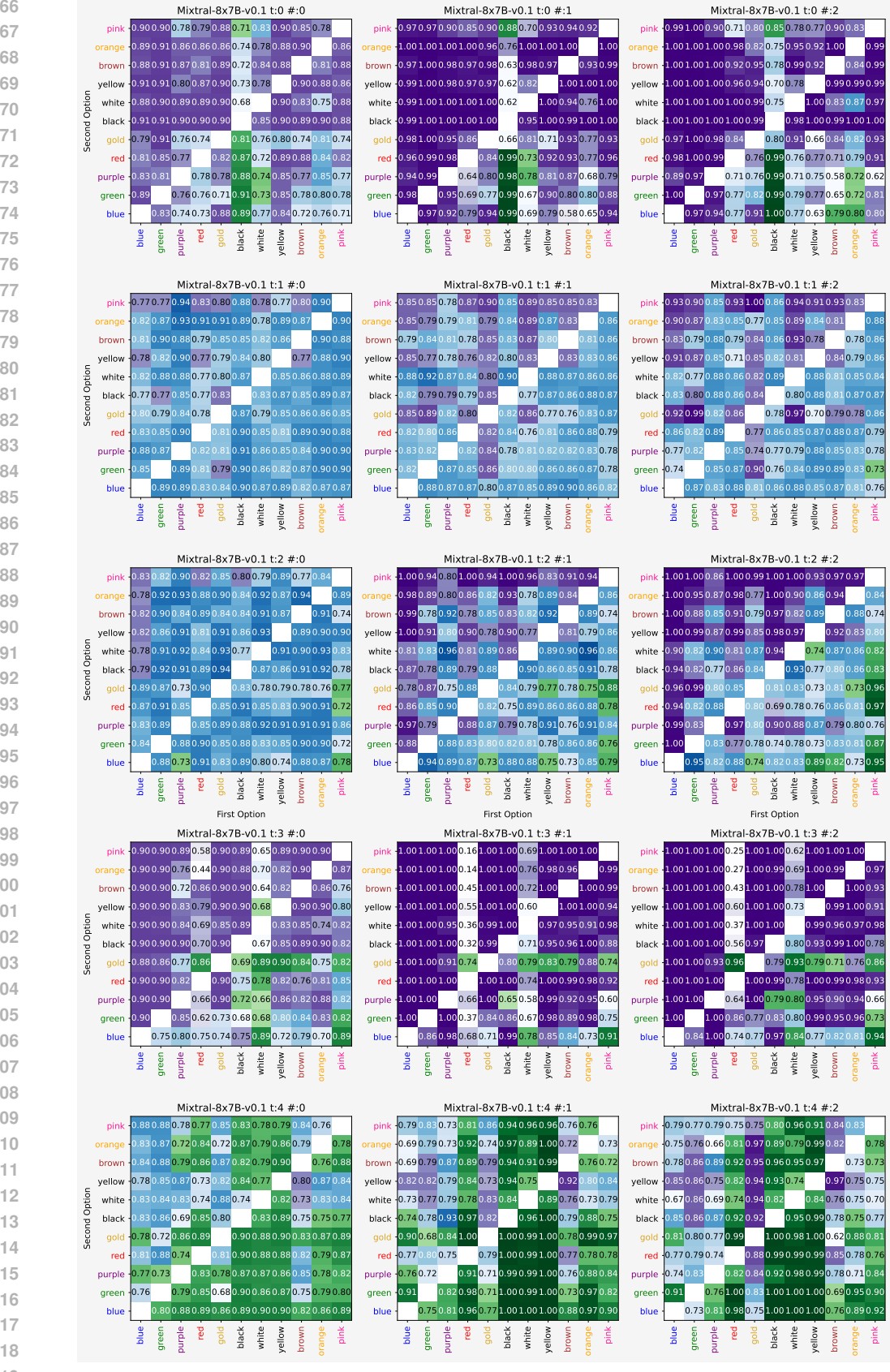

Figure 21: **Model Behavior Profile: mixtral-8x7b-v0.1 (chat).**

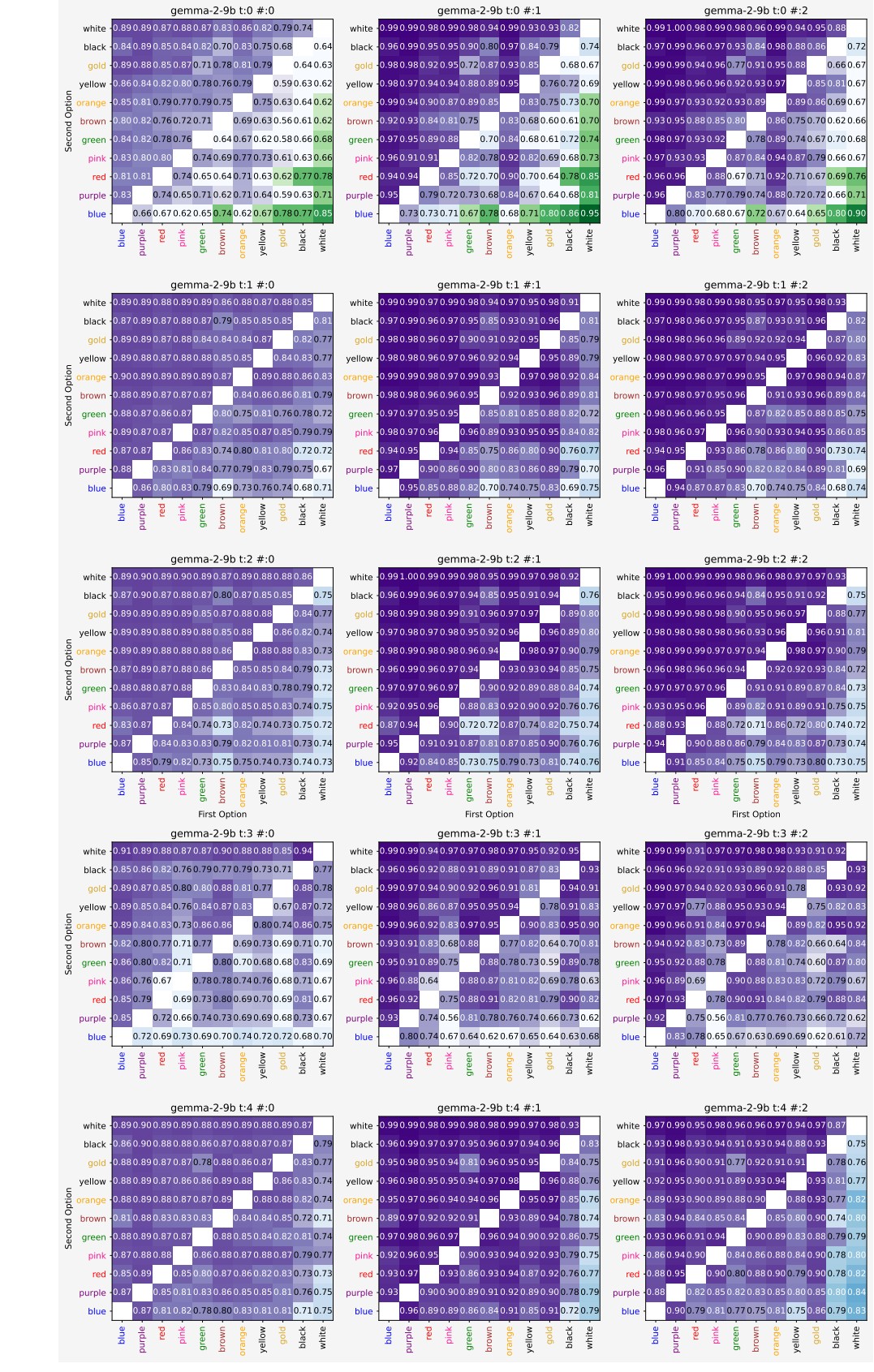

Figure 22: **Model Behavior Profile: gemma-2-9b (chat).**

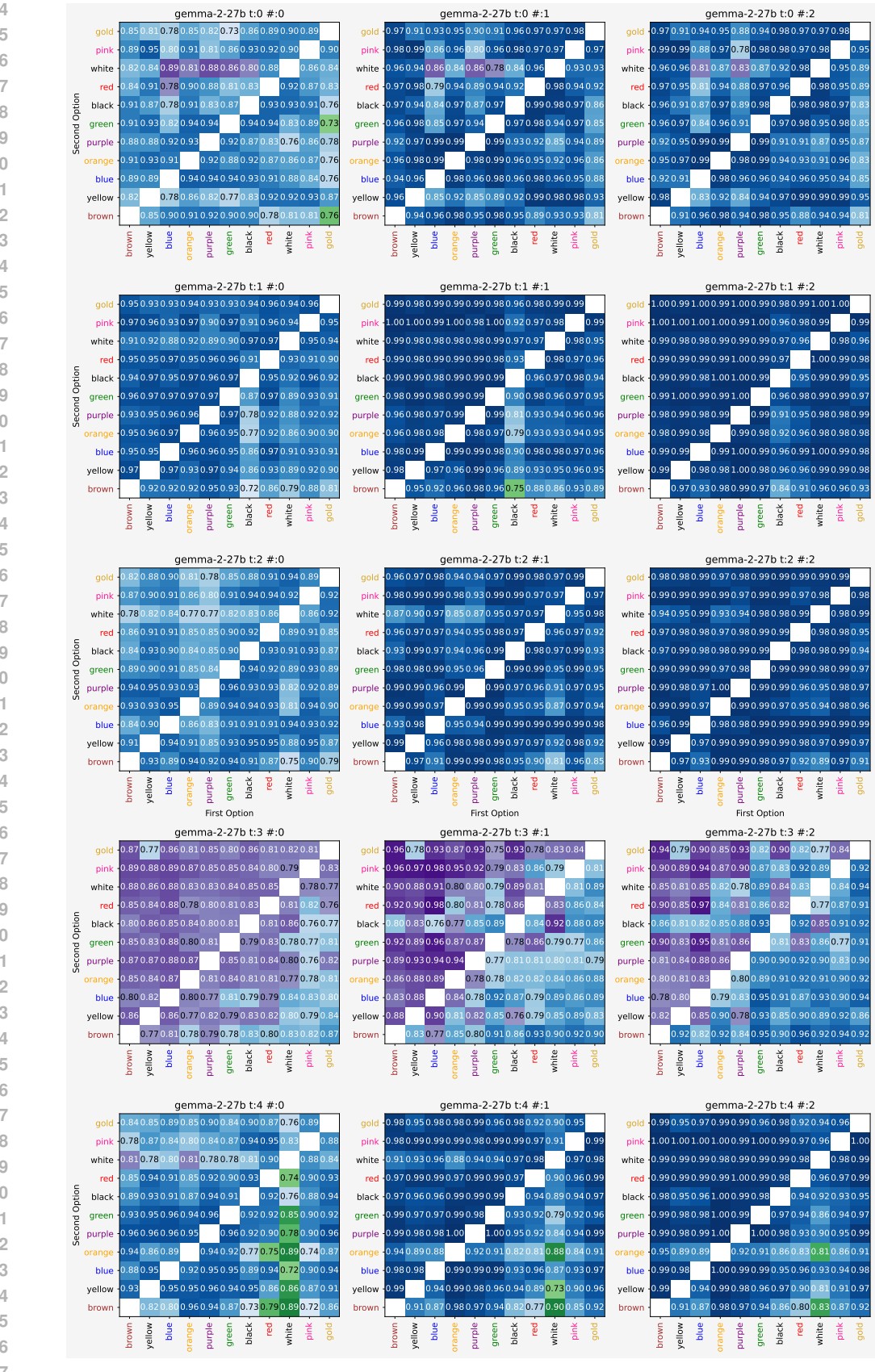

Figure 23: **Model Behavior Profile: gemma-2-27b (chat).**

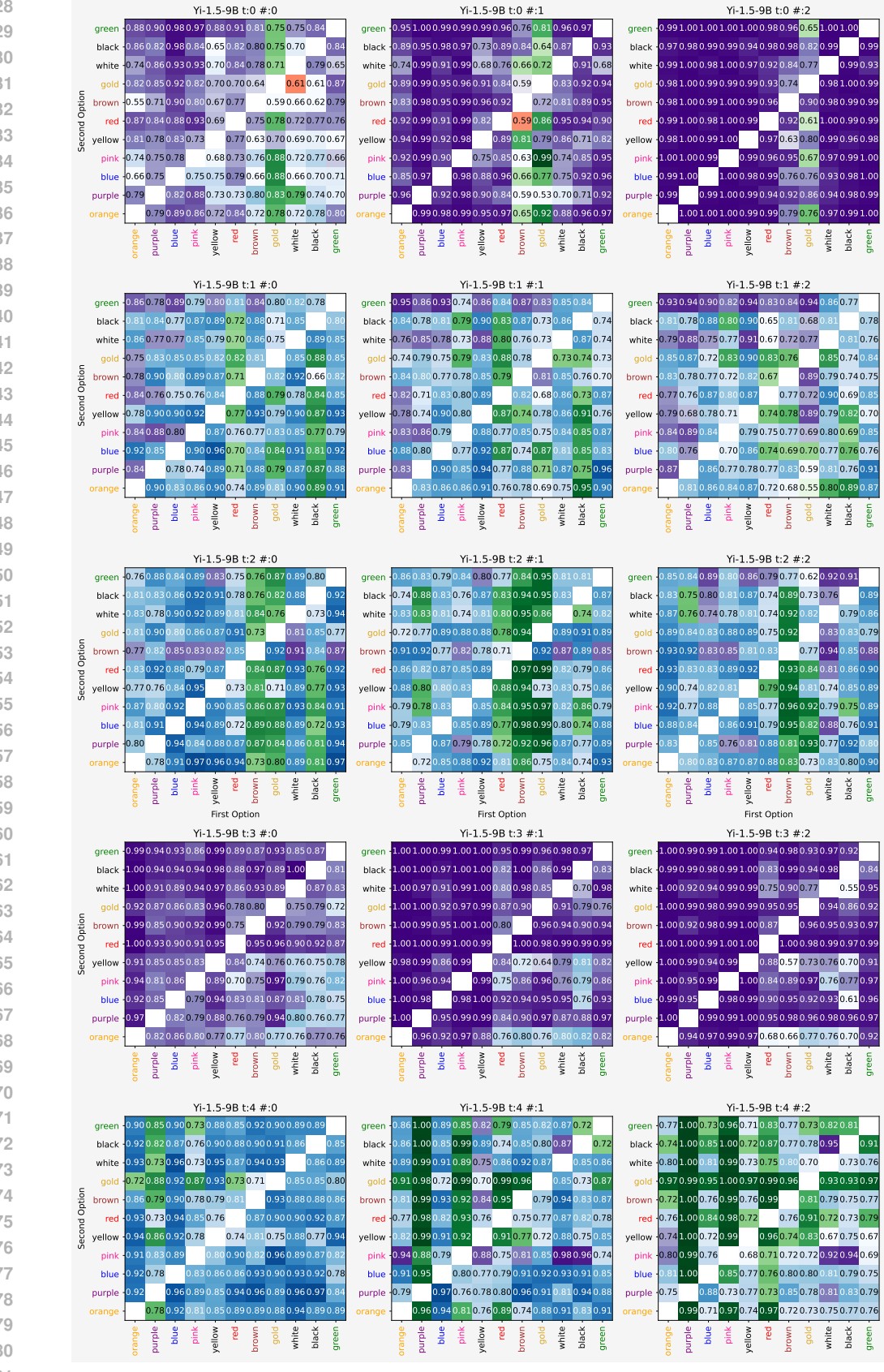

Figure 24: **Model Behavior Profile: Yi-1.5-9B (chat).**

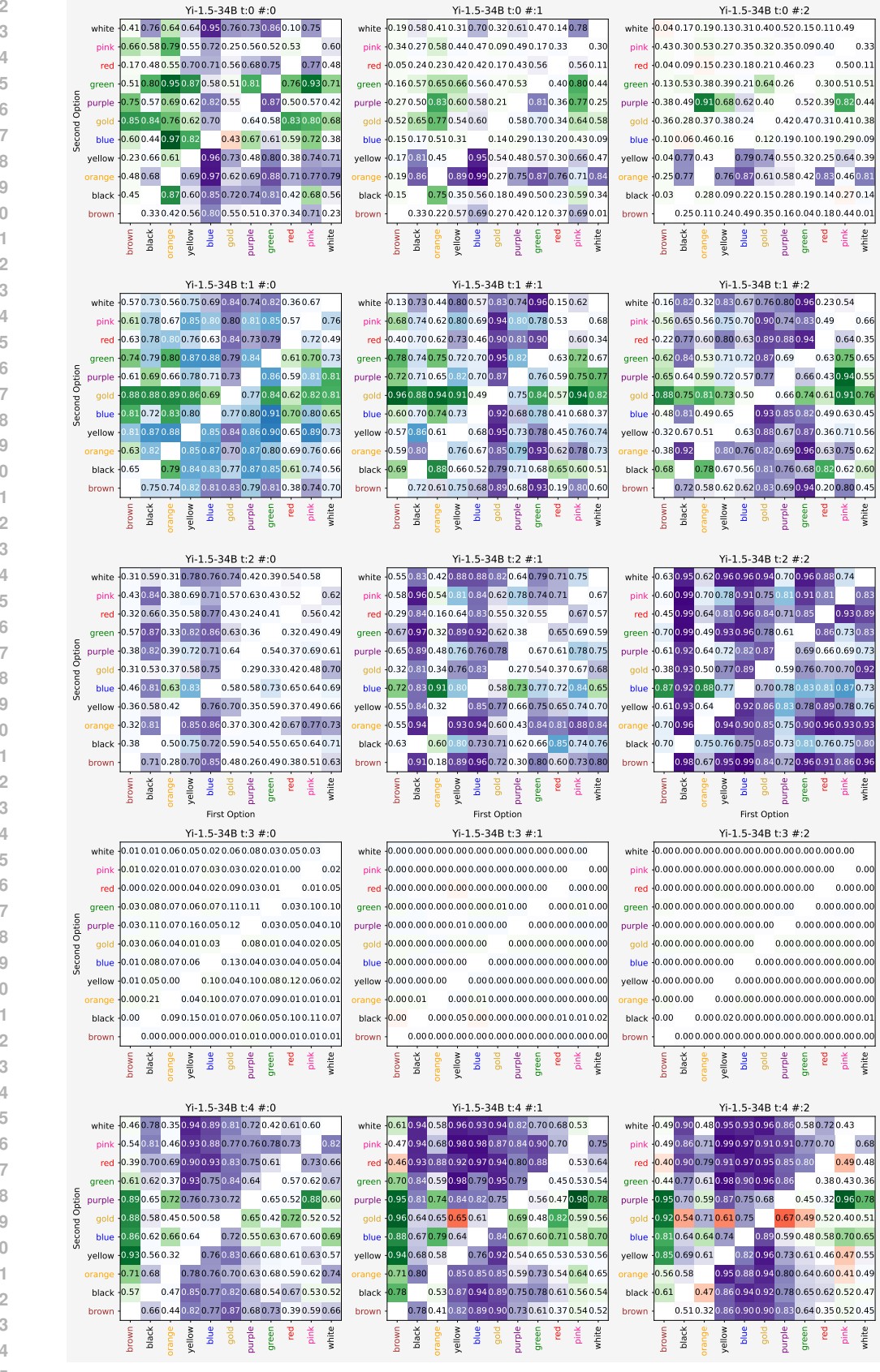

Figure 25: **Model Behavior Profile: Yi-1.5-34B (chat).**

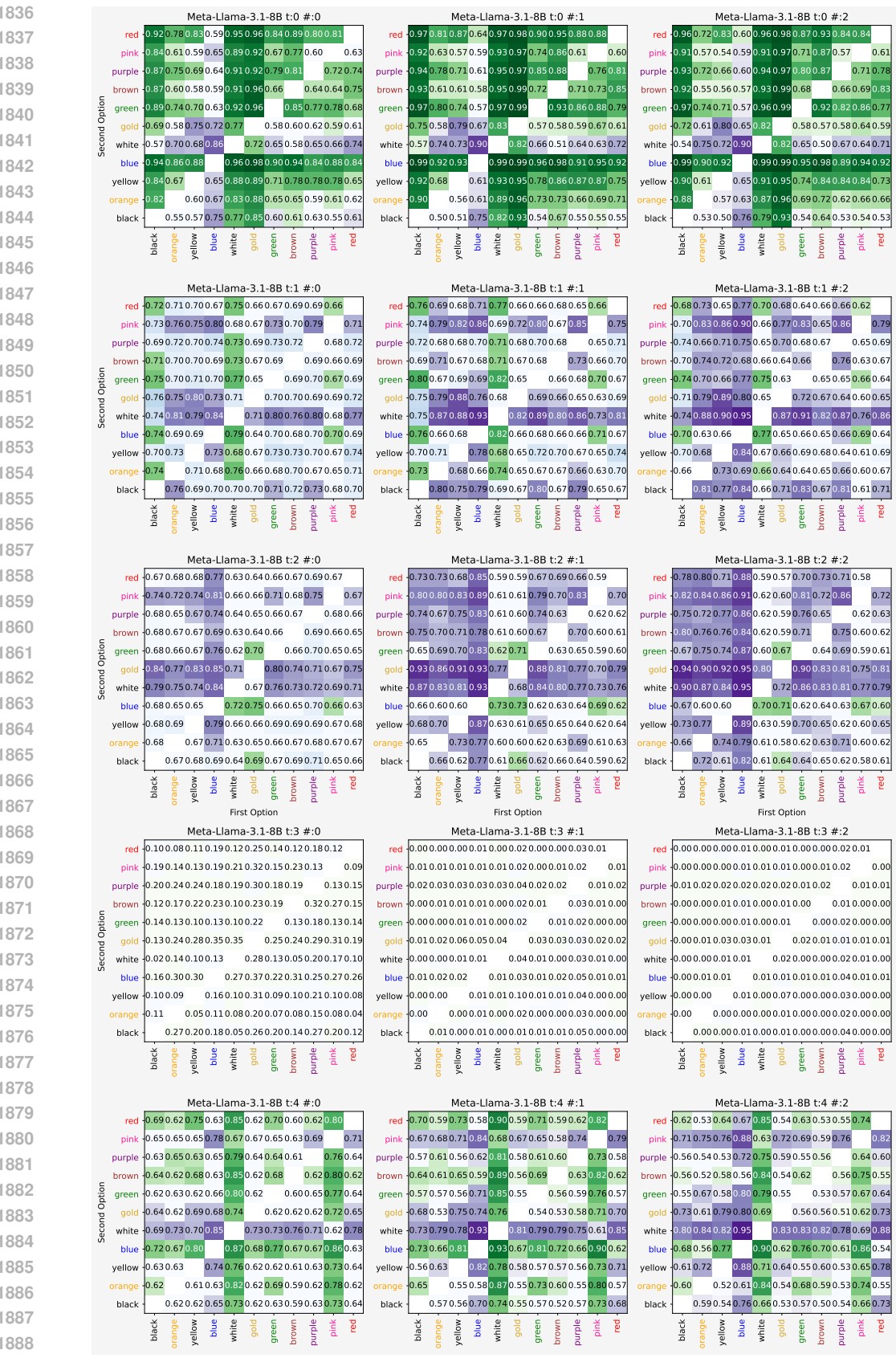

Figure 26: **Model Behavior Profile: llama-3.1-8b (chat).**

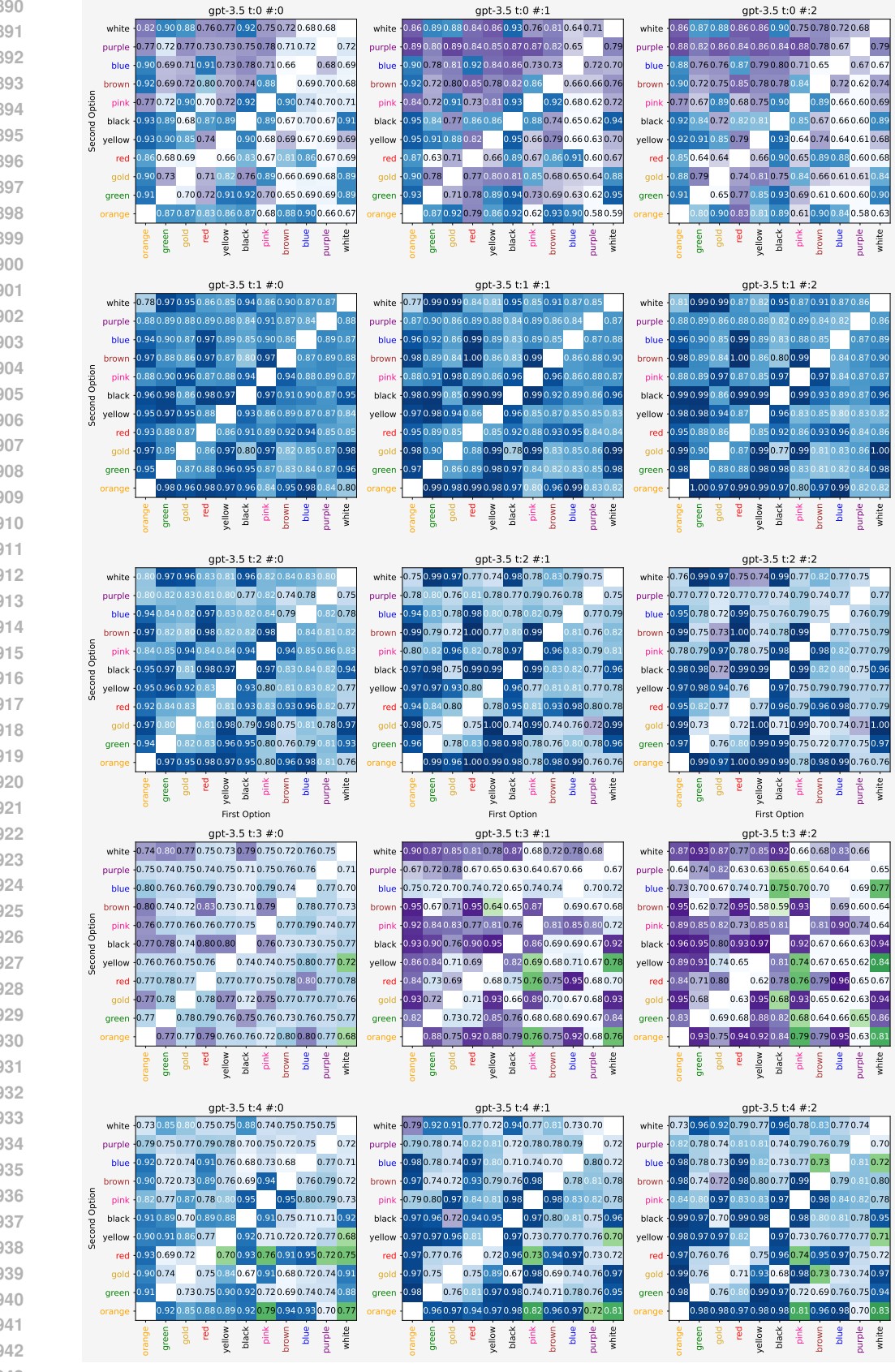

Figure 27: **Model Behavior Profile: gpt-3.5 (chat).**

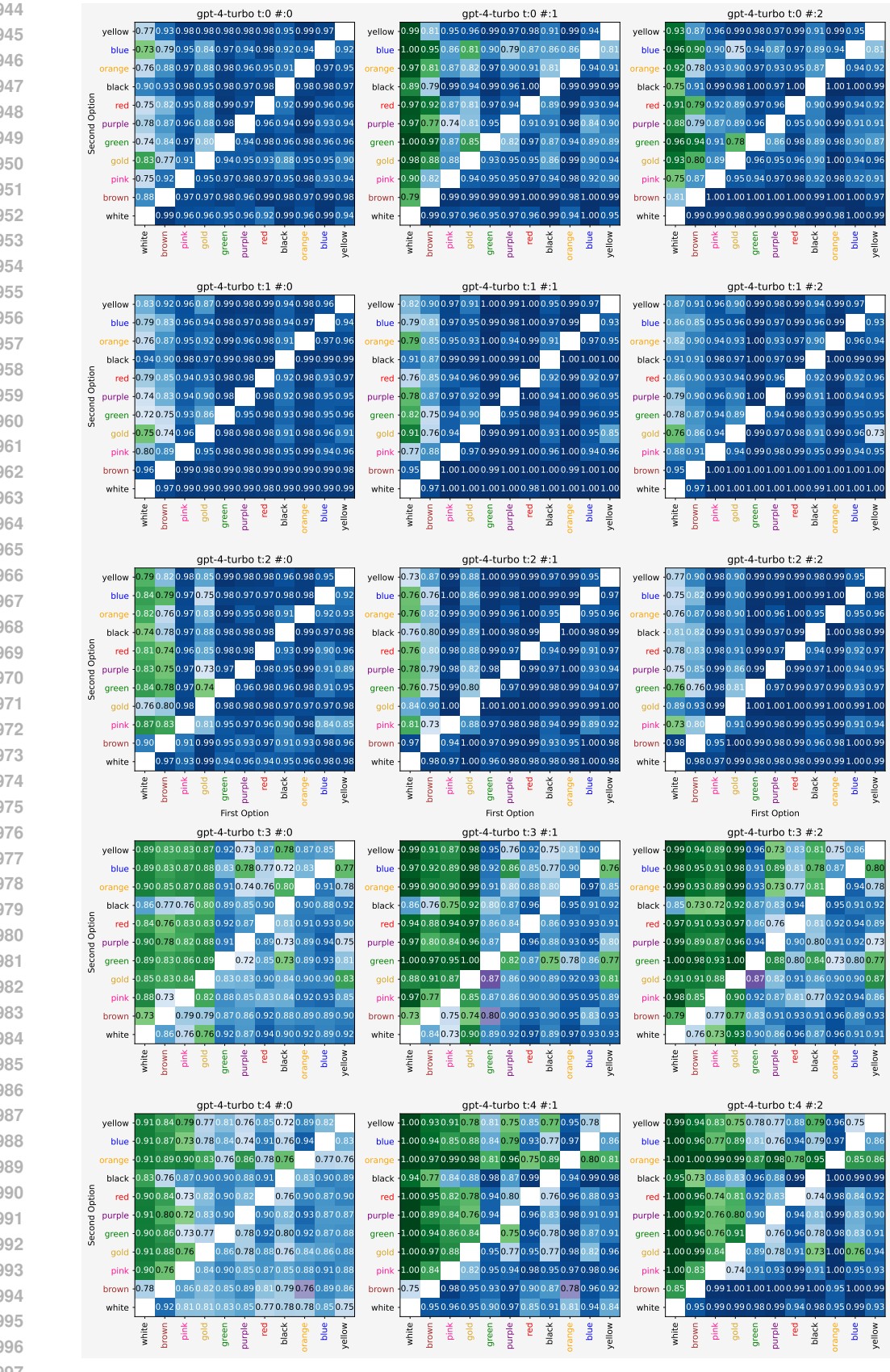

Figure 28: **Model Behavior Profile: gpt-4-turbo (chat).**

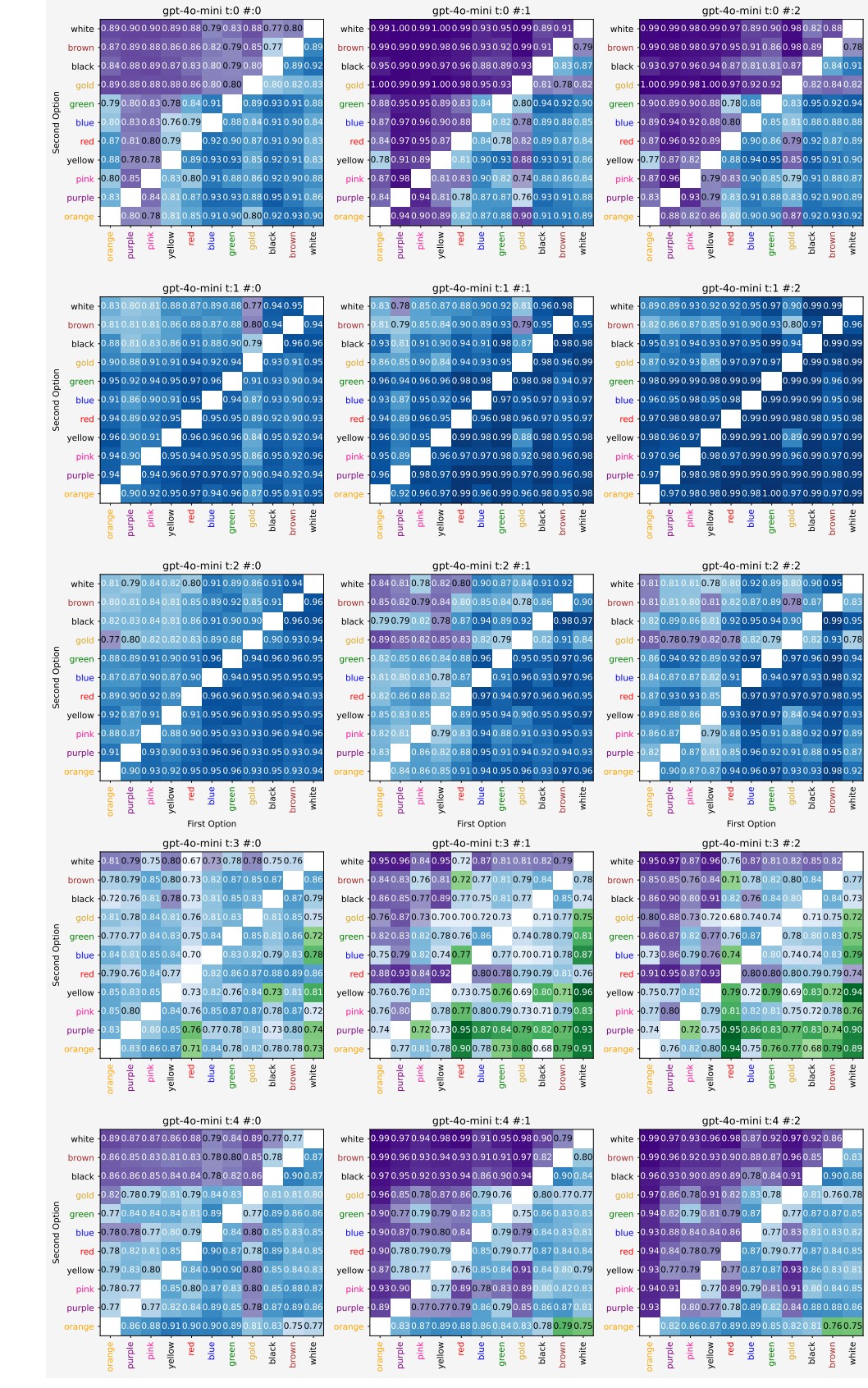

Figure 29: **Model Behavior Profile: gpt-4o-mini (chat).**

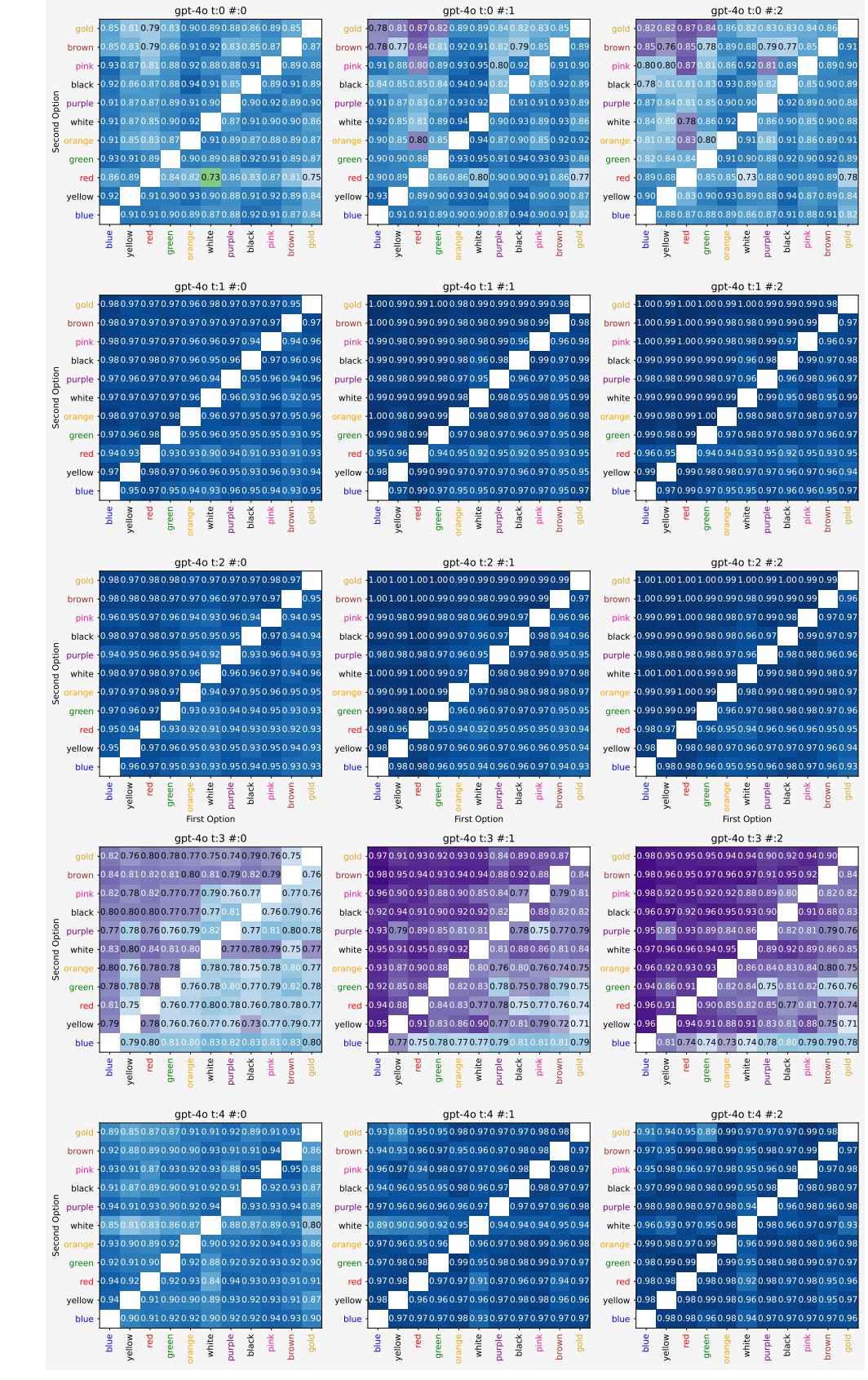

Figure 30: **Model Behavior Profile: gpt-4o (chat).**

