# OpenReview forum: "Are Language Model Logits Calibrated?"
_ICLR.cc/2025/Conference — ICLR 2025 Conference Withdrawn Submission_

### Official Review · Reviewer_u7uC · 2024-10-31

**Soundness:** 2
**Presentation:** 3
**Contribution:** 2
**Rating:** 3
**Confidence:** 4

**Summary:**

This paper focuses on the calibration of language models. It defines calibration as the degree to which the output distribution probabilities of the candidate tokens are aligned to the likelihood inferred from the context.
The author evaluated the case of drawing marbles on open and close-source models and showed that LMs are poorly calibrated in this case.
The author also conducted further analysis and a human study to compare the calibration level of the model to humans.

**Strengths:**

1. This paper is clearly written and easy to read.
2. The idea of using context as the implicit likelihood for calibration is interesting and provides a good starting point for analysing the calibration level of language models.
3. The human study is a good addition to show that both human and language models are biased, motivating the research for improving the calibration level of language models.

**Weaknesses:**

1. The main and critical issue of this paper is that they didn't discuss the impact of the temperature of the softmax operation for getting the probabilities. The choice of temperature can greatly influence the entropy and final calibration level of the model, which will greatly affect the conclusion drawn from the paper. As shown in Table 1, changing temperature from 0.1 to 1.0 greatly changed the performance of the random baseline.

2. The overall contribution of the paper is poor. The issue of low entropy of the Chat models and their token bias has been shown in different papers. The novelty of this paper is primarily on the definition of the calibration by using context information.

3. The author only focuses on the next token position. It is also possible that the model may not give you the answer in the text token. As discussed in the paper, the PM can be low, meaning all valid tokens have low probabilities. If the model outputs 'of red' rather than 'red', probabilities of the second position should also be considered.

**Questions:**

1. What would the result be like if you search over different temperature choices?

2. Have you checked the actual text output of the models if they directly give the valid token in the next token position?

---

### Official Review · Reviewer_LhL4 · 2024-11-01

**Soundness:** 4
**Presentation:** 4
**Contribution:** 2
**Rating:** 5
**Confidence:** 5

**Summary:**

The authors find that LLM predictions are poorly calibrated for outputs that should ideally recover some aleatoric uncertainty, like the probability of drawing marbles of one color of out of a bag. Different models behave in different systematically uncalibrated ways, and sota models often allocate all of the probability to just one token.

**Strengths:**

- Well defined problem statement; the authors don't make any claims that they can't justify.
- Thorough set of experiments to validate the existence of this problem, with very detailed methodology.
- A human study that shows, interestingly, that humans are also not well-calibrated and that by some measures, the models are actually better calibrated than humans.

**Weaknesses:**

- There seems to be a lot of relevant work on LLMs expressing uncertainty that could be cited here (e.g., see https://arxiv.org/abs/2401.06730 and the related work discussed therein).
- There is no discussion of conformal prediction and how it might be used to address this problem, it it hasn't been done so already.
- In summary, the paper identifies a problem that is already well known in some capacity---albeit maybe not in the particular formulation proposed here. Although the paper is well-written and the problem is well-articulated, no real solution is offered. Because of this, I lean towards 'borderline reject' for an ICLR paper, though I could seeing it faring better at an NLP or CL conference.

**Questions:**

n/a

---

### Official Review · Reviewer_xEc3 · 2024-11-04

**Soundness:** 3
**Presentation:** 3
**Contribution:** 1
**Rating:** 5
**Confidence:** 3

**Summary:**

This paper analyses how calibrated language model (LM) outputs are when prompted with explicit probabilistic information (e.g., prompt: “an urn has 5 blue and 3 red balls, I pick one, this ball is”, LM output: blue/red”).
They perform two main experiments:
* “Distributions”. In this first experiment, they prompt the model with information about a uniform distribution (e.g., “Sampling from a uniform distribution with support from 2 inclusive to 5 exclusive, I sampled the number”) and evaluate the probability a LM places on tokens $\{0, 1, 2, 3, …, 9\}$.
* “Probabilities”. In this second experiment, they prompt the model with an implicit distribution (e.g., “From 17 red marbles and 99 blue marbles, Billy reached blindly into the bag and grabbed a marble with the colour”) and see whether the model places calibrated probability on the tokens of interest (in this case, $\{red, blue\}$).

In the first experiment, the paper finds that instruction-tuned models are *less* calibrated than base models. It also finds that models have systematic preferences for some tokens (e.g., systematically preferring token 3 to others). In the second experiment, the paper finds that instruction-tuned models are *more* calibrated than base models (the opposite from the first experiment).

They also perform a human study, where they compare LM to human behaviour.

**Strengths:**

This paper investigates an interesting question: whether language models produce calibrated outputs when prompted with explicit probabilistic information.

The paper also proposes two related experiments to investigate model behaviour in this setting, analysing results in these two settings in detail.

**Weaknesses:**

While the paper performs two interesting experiments to investigate the role of explicit probabilistic calibration on language models, I believe that in its current state, the insights that can be drawn from it are somewhat limited. The paper performs a behavioural analysis on language models with two relatively simple and similar settings. I’d expect an ICLR paper to perform a more thorough analysis. Some suggestions below.

I think the paper would be much stronger if it performed one of the following analysis (or both):
* *Training data analysis.* While this may not be possible with the models currently analysed in the paper, the authors could extend their analysis to, e.g., Pythia or OlMO, and evaluate how the analysed LM behaviour relates to different statistics of the training data. The paper speculates about this in section 6, but this could be analysed.
* *Mechanistic analysis.* The paper currently evaluates model behaviour purely as a black box. Performing mechanistic/causal analysis of how this behaviour relates to model internal activations could make the results more insightful. E.g., the paper could use distributed alignment search (Geiger et al. 2023) to find subspaces in the model which control the model’s behaviour on these tasks.


Geiger et al. 2023. Finding Alignments Between Interpretable Causal Variables and Distributed Neural Representations

**Questions:**

> Title: Are Language Model Logits Calibrated

The word logits is not used in the paper at the moment and logits (the pre-softmax activations) are not analysed---only the probabilities output by LMs (post-softmax) are analysed. I thus believe the word "logits" doesn’t belong in the paper’s title. Further, the title—in my opinion—suggests the paper is about a more traditional notion of model calibration (e.g., Expected Calibration Error; ECE). Changing the title to highlight this different “view” of model calibration could be helpful. E.g.,  “Are Language Model Outputs Calibrated to Explicit Probabilistic Prompts” or something analogous.

> Most Plots

The Figures in this paper are not readable when printed in black and white.

---

### Official Review · Reviewer_Aaqb · 2024-11-06

**Soundness:** 3
**Presentation:** 4
**Contribution:** 2
**Rating:** 5
**Confidence:** 4

**Summary:**

The study investigates the alignment of language models’ output probabilities with the numeric or probabilistic information in the contexts they’re given. They explore models that have undergone different fine-tuning techniques (instruction tuning, preference alignment) and see how this affects the model’s explicit reasoning in comparison to base models. They look at whether biases in token probabilities (e.g, a first-mentioned bias) can be identities. They find that across model architectures, language models are generally not well calibrated in this respect. Instruction-tuning seems to exacerbate the issue, often leading to mode-collapse. They also observe some interesting systematic biases for different model families/fine-tuning strategies. These findings highlight an important limitation of language models.

**Strengths:**

* The work points out an important shortcoming of language models, namely, an inability to accurately reflect probabilistic information given in a context. They further show how recent fine-tuning approaches affect this calibration
* Many different models are evaluated, including the base and chat versions of several popular architectures, showing consistency and their observations and allowing for broader conclusions
* The paper is well-written and the authors provide good motivation for the problem they’re exploring

**Weaknesses:**

* The contributions of the work are rather limited. A very specific question is being asked and its unclear how relevant this question is to the broader community. While the work reveals an undesirable model behavior, it doesn’t propose methods for fixing these behaviors. There is a short discussion of potential reasons for the behaviors, but no empirical evidence for or against these hypotheses are given
* Many of the results are missing standard errors or significance tests.
* The enormous set of results in the appendix is difficult to navigate

**Questions:**

* Could you provide more details about the datasets that you’re using? For example, how many datapoints are in each and how “diverse” are the different prompts
* In 3.2 in the definition of PM(T), should it be lowercase pi?

---

### Note · Authors · 2024-11-26

**Comment:**

Thanks again to the reviewers for their careful work.

**Withdrawal Confirmation:**

I have read and agree with the venue's withdrawal policy on behalf of myself and my co-authors.